# Seed structure and phosphorylation in the fuzzy coat impact tau seeding competency

Alysa Kasen[1], Sofia Lövestam[2], Libby Breton[1], Lindsay Meyerdirk[1], Jacob Alec McPhail[3,4], Kristin Piche [3], Ariel Louwrier[3], Colt D. Capan[1,5], Hyoungjoo Lee[1,5], Sjors H. W. Scheres [2] & Michael X. Henderson [1]✉

Tau misfolding into β-sheet–rich filaments and subsequent recruitment of monomeric tau are central to Alzheimer's disease (AD) pathogenesis. While cryo-EM has resolved the conformation of the AD tau core, the structural features conferring biological activity remain unclear. Here, we investigated how tau filament core structure and post-translational modifications influence seeding capacity in neurons and mice. Our findings show that although filament structure impacts seeding, the AD tau core alone is insufficient to fully replicate AD tau's biological activity. The unstructured fuzzy coat, particularly phosphorylation within this region, is essential for full seeding competence. Importantly, recombinant tau filaments bearing twelve phospho-mimetic residues (PAD12 tau) and adopting the AD fold recapitulate the seeding activity of native AD tau. These results demonstrate that tau filament pathogenicity arises from the combined contributions of both the ordered core structure and post-translational modifications within the fuzzy coat, providing critical insights into mechanisms underlying tau-driven neurodegeneration.

Alzheimer's disease (AD) is the most common cause of dementia and is characterized pathologically by misfolded aggregates of extracellular amyloid β (Aβ) plaques and intracellular tau tangles[1]. While Aβ plaques accumulate throughout the brain with age, many individuals with mild plaque burden do not develop dementia. The mechanisms that link aggregation of Aβ with the aggregation of tau are still unclear, but tau misfolding correlates more directly to disease onset and progression of dementia[2].

Tau, encoded by the *MAPT* gene, is a natively unfolded protein. It can bind microtubules where it adopts secondary structure and modulates microtubule stability and trafficking[3–5]. Tau undergoes extensive post-translational modification (PTM), and many PTMs regulate tau function[3,6]. In particular, phosphorylation in the microtubule binding repeats causes tau to dissociate from microtubules[7,8]. Phosphorylation of tau outside of the microtubule binding repeats has been linked to its assembly into amyloid filaments, which can serve as a

template for the misfolding of additional tau monomers into the filaments as seen in AD brains[9–11].

AD tau filaments are the major constituent of pathological tangles and are modified by several PTMs, including phosphorylation, acetylation, and ubiquitination. Interestingly, phosphorylation of AD tau occurs most prominently in the proline-rich domain and the C-terminal domain, while ubiquitination is localized to the microtubule binding repeats, and acetylation occurs in the microtubule binding repeats and C-terminal domain[9]. Many of these modifications are thought to occur after filament assembly, and antibodies directed against specific modifications of tau have been used to stage the maturity of tau tangles[12–14].

While it is known that tau PTMs can influence the formation of filamentous aggregates, it is not clear if they are important in the further templated recruitment of tau into filaments. One clear influence on templated recruitment is the ordered core structure of the tau

[1]Department of Neurodegenerative Science, Van Andel Institute, Grand Rapids, MI, USA. [2]MRC Laboratory of Molecular Biology, Cambridge, UK. [3]StressMarq Biosciences Inc., Victoria, BC, Canada. [4]Institute for Neurodegenerative Disease, Weill Institute for Neurosciences, University of California, San Francisco, CA, USA. [5]Mass Spectrometry Core, Van Andel Institute, Grand Rapids, MI, USA. ✉e-mail: michael.henderson@vai.org

filaments. Recent advances in structural biology have led to a flurry of structures of amyloid filaments from various neurodegenerative diseases[15]. Specific tau folds characterize different tauopathies[12,16]. The different structures of tau filaments in different diseases suggest that filaments assemble in specific circumstances in each disease. The conserved structures of tau in different regions of an individual with a given disease supports a model where new filaments form by templated recruitment. However, it remains unclear how PTMs and the tau filament core structure modulate this pathogenic recruitment process.

Recent efforts to understand the process of tau filament formation have used recombinant protein or animal models. Recombinant full-length wildtype tau is highly soluble, and requires the addition of negatively charged co-factors, such as heparin or RNA, to promote its spontaneous assembly into amyloid filaments[17–20]. However, the structures of co-factor-induced filaments are distinct from those found in diseased brain[21–23]. Furthermore, these co-factor-induced filaments lack seeding activity in wildtype mice[24], whereas AD-derived tau shows potent seeding capacity[24]. We hypothesized that this difference in seeding capacity is related to the structure and/or PTMs on tau.

In the current study, we tested this hypothesis by assessing the seeding capacity of recombinant tau filaments with various structures[25]. Recombinant tau filaments with the AD structural core showed improved seeding capacity compared to filaments with different, non-disease-related structures, in both primary neurons and mice brains, but failed to capture the full seeding capacity of AD tau. These observations suggest that the structure of the ordered filament core matters for seeding, but that the tau "fuzzy coat", the disordered regions of tau on both the carboxy-terminal and amino-terminal sides of the ordered filament core (aa1–305, aa379-444), also plays a role in modulating seeding capacity. Indeed, protease removal of the fuzzy coat reduced tau seeding capacity. Moreover, phosphatase treatment of AD tau reduced tau seeding capacity to a similar extent as removal of the fuzzy coat, while re-phosphorylation of AD tau partially rescued seeding capacity. Finally, filaments of recombinant tau with twelve phospho-mimetic mutations (PAD12 tau) with the same structure as AD tau fully recapitulated the seeding capacity of AD tau. Together, these experiments support a model where a combination of tau filament core structure and phosphorylation in the fuzzy coat facilitate tau seeding.

## Results
### Recombinant tau filaments do not recapitulate the seeding capacity of AD PHFs

Several mouse models have been generated that develop hyperphosphorylated tau inclusions[26,27]. One of the most widely used mouse models overexpresses P301S mutant tau under the mouse prion protein promoter and develops age-dependent tau inclusions and dramatic hippocampal degeneration[28]. While these transgenic models are useful to examine modifiers of tau pathology and neuron death, they do not recapitulate the cell type specificity, progression pattern, or structure of inclusions present in AD[29,30]. To more directly assess pathogenicity of AD tau, seed-based models have been generated where sarkosyl-insoluble tau extracted from AD brains is directly inoculated into wildtype or transgenic mice, inducing a progressive and cell type-selective tauopathy[24,31]. However, these models require access to AD brain tissue since recombinant tau fibrils do not seed efficiently, and heterogeneity related to sample preparation is a potential variable.

To assess the seeding capacity of various tau filament preparations, we cultured primary cortical neurons from wildtype mice. While adult mice express only 4R (4 microtubule binding repeats) tau, fetal mouse neurons express both 3R and 4R tau (Fig. 1a)[32,33]. This assay relies on the endogenous capacity of primary neurons to generate aggregates after the addition of tau filaments. No aggregates form in

the absence of seeding, and we have previously found that seeding capacity in this system is a good predictor of seeding capacity in vivo. Neurons are treated with filaments for 7 days in vitro (DIV) and fixed and stained at 28 DIV (Fig. 1b). To avoid potential detection of the human tau seeds, soluble tau is extracted using 2% of the detergent HDTA and neurons are stained with the T49 antibody, which recognizes mouse, but not human, tau[34]. We demonstrated good seeding and detection of aggregates 21 days after treatment with 25 nM AD tau, but not after treatment with PBS (Fig. 1c).

We then proceeded to test different versions of tau filaments as seeds. We first assessed the seeding capacity of heparin-induced wildtype tau 2N4R fibrils (StressMarq Cat# SPR-480) or RNA-induced P301S tau 2N4R fibrils (StressMarq Cat# SPR-463). As a positive control, we extracted tau filaments from three neuropathologically confirmed AD cases that were pooled to minimize inter-case variability (Supplementary Fig. 1a–d). The final fraction of tau in the human brain-derived samples was 20.9% of total protein (Supplementary Fig. 1e, and Supplementary Fig. 1f). Recombinant tau filaments were added at a low dose of 25 nM, a medium dose of 125 nM, and a high dose of 500 nM. AD tau was only added to 25 nM and 125 nM, because the high dose induced cell death. Both types of full-length recombinant tau filaments seeded only minimal tau pathology in wildtype primary neurons (Fig. 1d, e). Heparin-induced filaments caused cell death at 125 nM; RNA-induced P301S filaments induced cell death at 500 nM (Fig. 1f). We hypothesized that the low seeding of these filaments could be due to the difference in their core structures compared to AD tau.

### Recombinant tau core with AD conformation has improved seeding capacity

Whereas full-length recombinant tau filaments that form in the presence of heparin or RNA adopt different structures compared to those observed from human brains[21,22], dGAE tau (residues 297-391) readily forms filaments with the AD tau structure without co-factors. A recent cryo-EM study characterized the structures of recombinant tau filaments that formed in 76 different assembly conditions, including different tau constructs, buffers and shaking speeds[25]. We hypothesized that filaments that adopt the same ordered core as in AD tau would have increased seeding capacity compared to other, non-disease associated structures. To test this hypothesis, we used three of the previously reported filament types (4a, 42ab, and 12a) in our seeding assay with primary cortical neurons, and we confirmed by cryo-EM that the preparations used as seeds indeed adopted the same structures as reported previously (Fig. 2a). In agreement with the previous report, only 4a filaments (dGAE construct, 200 mM $MgCl_2$, 200 rpm, 48 h) recapitulate the AD tau core structure. 12a filaments (dGAE construct, 200 µM $CuCl_2$, 200 rpm, 48 h) have a unique fold that does not recapitulate a known disease-associated structure. 42ab filaments, which were assembled from tau297-408 with four phosphomimetic mutations in similar conditions to 4a (200 mM $MgCl_2$, 200 rpm, 48 hours), contain two protofilaments with the AD tau fold, but with a different inter-protofilament packing.

We treated primary cortical neurons with each of the three recombinant filaments or with AD tau. Due to the difference in filament length between the recombinant filaments and AD tau, we added filaments at equimolar concentrations to normalize for the number of 'tau seeds'. We found that 4a filaments had a higher seeding capacity than 12a or 42ab filaments, again supporting the notion that the structure of the ordered core of tau seeds affects their seeding capacity. However, the seeding capacity of the 4a filaments was low compared to that of AD tau (Fig. 2b, c). To understand if this result relied on neuron type, we repeated the seeding assay in primary hippocampal neurons, which have a higher proportion of excitatory neurons. We found a similar difference in the seeding capacity of the filaments, with 4a filaments being the most potent of the recombinant tau filaments (Supplementary Fig. 2).

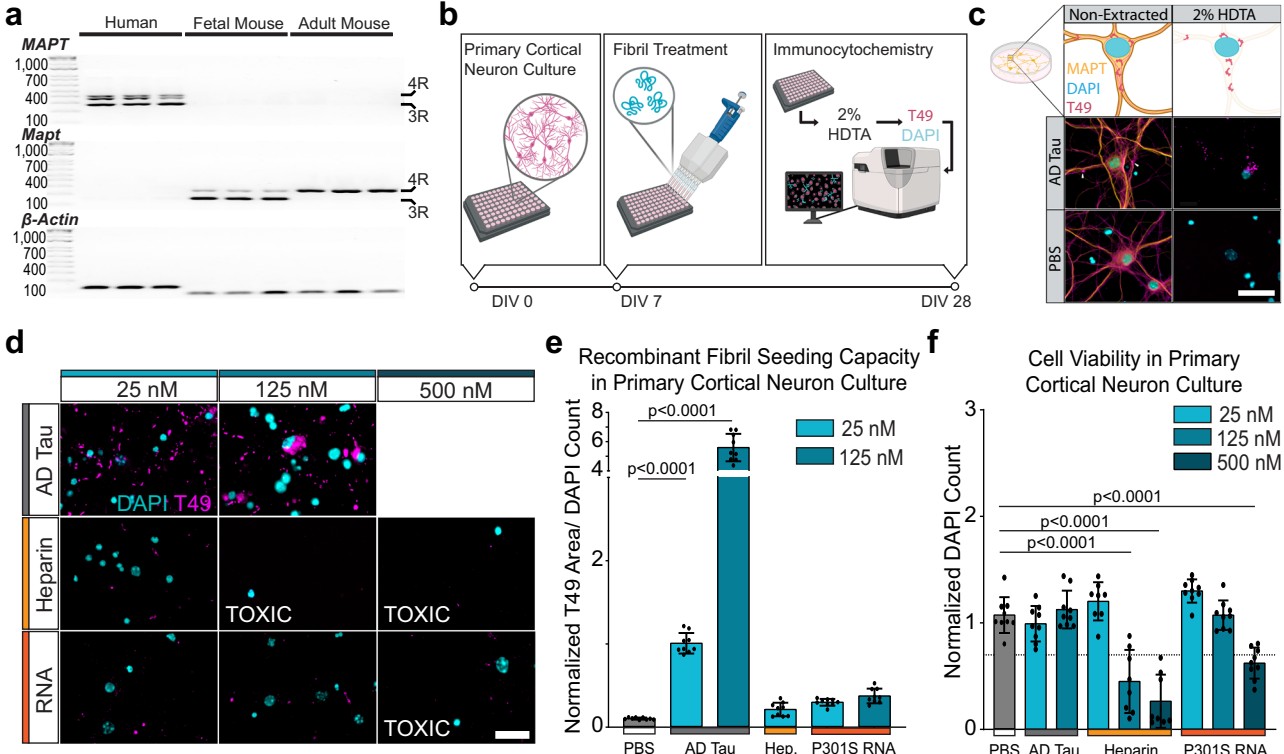

**Fig. 1 | Recombinant tau filaments do not recapitulate the seeding capacity of AD PHFs. a** PCR products of human, fetal mouse, or wildtype adult mouse cortical brain tissue for human tau (*MAPT*), murine tau (*Mapt*), and β-actin. N = 3 independent brain extracts. Units in BPs. **b** Experimental schematic. Primary neurons were cultured, treated with sonicated tau filaments at 7 DIV, and maintained until 28 DIV when the soluble protein was extracted with 2% HDTA, and neurons were fixed and stained. Schematic created in BioRender. Kasen, A. (https://BioRender. com/h8m20ur). **c** Schematic of representative images showing the removal of soluble protein extraction by 2% HDTA to detect insoluble mouse tau with T49. Scale bar = 30 µm. Schematic created in BioRender. Kasen, A. (https://BioRender. com/h8m20ur). **d** Representative insoluble tau staining across filament type and molar concentration. Scale bar = 25 µm. Treatments that resulted in cell death are annotated as TOXIC. **e** Insoluble tau pathology (T49) was quantified relative to DAPI count. Data is presented as mean ± SEM with individual values plotted. N = 9 independent wells from 3 separate cultures. **** *p* < 0.0001, One-way Welch ANOVA test and Dunnett's T3 multiple comparison test compared to PBS control. **f** Cell viability at DIV 28 measured by normalized DAPI count normalized to AD Tau 25 nM treatment. Data is presented as mean ± SEM with individual values plotted. The line at y = 0.7 represents 30% cell death and considered a toxic dose. N = 9 independent wells from 3 separate cultures. ****p < 0.0001, One-way Welch ANOVA test and Dunnett's T3 multiple comparison test compared to PBS control. Source data are provided as a Source Data file.

## The AD PHF core alone does not replicate seeding capacity in vivo

We and others have previously demonstrated that the injection of AD tau into the hippocampus of wildtype mice leads to the seeded aggregation of endogenous mouse tau[24,35]. Seeded aggregation is detectable by immunohistochemistry as early as 1 month post injection (MPI) and shows progressive increases in pathology from 1 to 9 MPI, primarily in regions directly connected to the hippocampus[24,35]. Due to the enhanced seeding of 4a filaments in primary neurons, we also assessed the seeding capacity of recombinant tau filaments in vivo in wildtype mice. As a positive control, we injected wildtype mice with 2 µg AD tau in the hippocampus. Due to the lower seeding capacity of recombinant filaments in primary neurons, mice were injected with a higher amount (4 µg) of 4a, 42ab, or 12a filaments (Fig. 3a). Imaging by negative stain transmission electron microscopy (TEM), supported by dynamic light scattering analyses, confirmed that, after sonication, both AD tau seeds and the three types of recombinant tau seeds comprise filaments of lengths that have been reported to be effective for tau seeding (1 nm to 250 nm)[36,37] (Supplementary Fig. 3).

At 9 MPI, tissues were sectioned and stained for pS202/T205 (AT8), a marker of hyperphosphorylated, aggregated tau. As expected, AD tau induced robust hyperphosphorylated tau pathology in the hippocampus, entorhinal cortex, and supramammillary nucleus (Fig. 3b). In contrast, 42ab and 12a filaments did not induce detectable tau pathology. The 4a tau filaments induced a low amount of

pathology, with rare inclusions observed in the hilus of the hippocampus and supramammillary nucleus. We were surprised that recombinant tau with the AD core filament structure, even at a higher molar concentration, was unable to seed tau pathology to the same extent as AD tau. Given the partial seeding of 4a at 9 MPI, we hypothesized that neurons were not susceptible to seeding with these filaments, or that seeding with the recombinant filaments was much slower. To discern between these two possibilities, an additional cohort of mice was injected with 4a filaments and aged to 18 MPI. Seeding by 4a filaments was more pronounced at 18 MPI, with several neurons in the hilus and supramammillary nucleus showing extensive cell body and neuritic pathology (Fig. 3b, c). Together, these results suggest that seeding with 4a filaments was delayed compared to seeding with AD tau.

## Mice with 3R and 4R tau show increased seeding capacity

The amino acid sequence in the ordered core of AD tau filaments is identical in mice and humans[38]. However, adult wildtype mice only express 4R tau (Fig. 4a), whereas AD tau is composed of both 3R and 4R tau isoforms[38]. We hypothesized that the lack of 3R tau in adult mice may limit the pathology induced by the AD tau filaments. To test this hypothesis, we used *MAPT* knock-in (KI) mice in which the entire murine *Mapt* gene encoding tau is replaced with the human ortholog *MAPT*[34]. This model expresses all six isoforms of human tau, but in the same cells and with the same expression level as wildtype mice[34]. We

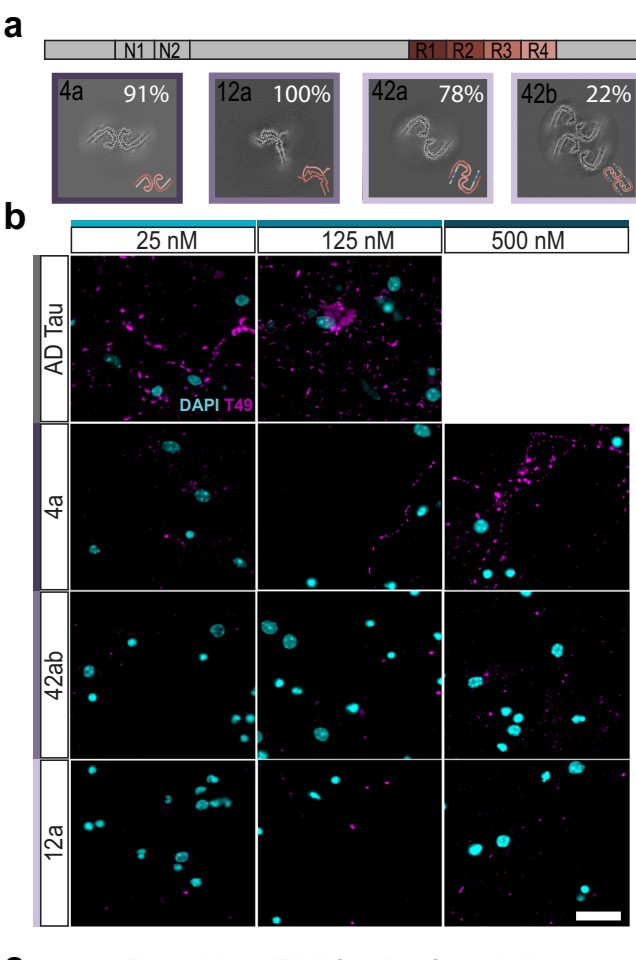

**a**

**b**

**c**

Recombinant Fibril Seeding Capacity in Primary Cortical Neuron Culture

**Fig. 2 | Recombinant tau core with AD conformation has improved seeding efficiency. a** Cryo-EM averages of 4a, 12a, and 42ab recombinant tau filaments used in this study. The percentage of fibril representative of the given structure is given for each structure. Cartoon schematic of recombinant filaments are shown in the bottom right corner of each box, with the colors corresponding to major domains of tau, shown above the filaments. **b** Representative images of primary cortical neurons treated with sonicated tau filaments at 3 doses and maintained in culture for 21 days after filament treatment. Scale bar = 20 μm. **c** Tau pathology, measured by T49, quantified relative to DAPI count. Data is normalized to 25 nM AD tau is presented as mean ± SEM with individual values plotted. N = 9 independent wells from 3 separate cultures. ****$p < 0.0001$, One-way Welch ANOVA test and Dunnett's T3 multiple comparison test compared to PBS control. Source data are provided as a Source Data file.

in the entorhinal cortex, hippocampus, and supramammillary nucleus. In contrast, 42ab filaments induced only sparse neuritic pathology, and no pathology was observed in mice treated with 12a filaments. Our observation that MAPT KI mice show improved seeding of both AD tau and recombinant tau filaments suggests that a lack of 3R tau isoforms in adult wildtype mice limits tau seeding efficiency. This suggests that cellular tau isoform composition may be an important factor in determining cellular vulnerability to specific tau strains in human disease.

To understand the quantitative burden of pathology in different animals, we implemented an established segmentation and brain registration strategy to quantify pathology burden in 318 regions of the midbrain[39,40] (Supplementary Fig. 4). While less total pathology was induced by 4a filaments compared to AD tau (Fig. 4d, e), the same regions were impacted by both types of seeds (Fig. 4f), suggesting that 4a can affect similar cell types, but with slower kinetics than AD tau.

### Loss of phosphorylation or fuzzy coat reduces tau seeding capacity

The 4a recombinant tau filaments lack a fuzzy coat. To explore whether the fuzzy coat contributes to tau seeding, we first removed it from AD tau filaments with pronase using an established protocol[41,42] (Fig. 5a). Cryo-EM has shown that pronase treatment of AD tau does not change the core structure of filaments[41]. Following pronase treatment, cleaved AD tau was centrifuged, and the pellet was washed twice with PBS to remove pronase and cleaved tau fragments (Supplementary Fig. 5a, and Supplementary Fig. 5b). Using antibodies against epitopes in the AD core and in the fuzzy coat, we confirmed that pronase treatment removed the amino-terminal and carboxyl-terminal parts of the fuzzy coat (Fig. 5b). The tau core antibody (OST00329W), which recognizes sites 323-363 in the processed tau protein, was used to calculate the abundance of tau by western blotting before and after pronase treatment to ensure equivalent amounts of tau in subsequent experiments. We further verified by negative stain TEM that tau filaments remained intact and had a similar width and pitch compared to AD tau prior to pronase treatment (Fig. 5c).

We also treated AD tau filaments with Lamda phosphatase[43] to determine whether phosphorylation of tau affects its seeding capacity. Following dephosphorylation, phosphatase was removed from the seeds by repeated centrifugation (Fig. 5d; and Supplementary Fig. 4a,c). We validated by western blot that phosphatase treatment removed 80-90% of the phosphorylation sites S202/T205 (AT8) and S396/S404 (PHF1), while phosphorylation at S262, which is inside the microtubule binding domain was minimally impacted (Fig. 5e). Similar to pronase treatment, phosphatase treatment preserved the overall appearance of AD tau filaments by negative stain TEM (Fig. 5f).

We then seeded primary cortical neurons with equimolar amounts of pronase-cleaved, phosphatase-treated, or untreated AD tau (Fig. 5g). Pronase or phosphatase treatment of AD tau resulted in a more than 4-fold reduction in seeding capacity (Fig. 5h). Notably, the

found that indeed 3R and 4R tau expression in these mice resembled that of the human brain (Fig. 4a).

We injected *MAPT* KI mice with 2 μg AD tau or 4 μg 4a, 42ab, or 12a filaments, and aged them to 9 MPI, as was previously done with wild-type mice. *MAPT* KI mice injected with AD tau showed elevated seeding of tau pathology compared to wildtype mice in the hippocampus and connected regions (Fig. 4b, c), similar to previous reports[34]. We observed that most of this increase was in neuritic pathology, with the hippocampal neuropil becoming filled with neuritic tau pathology. Seeding by 4a filaments was also elevated, and showed tau pathology

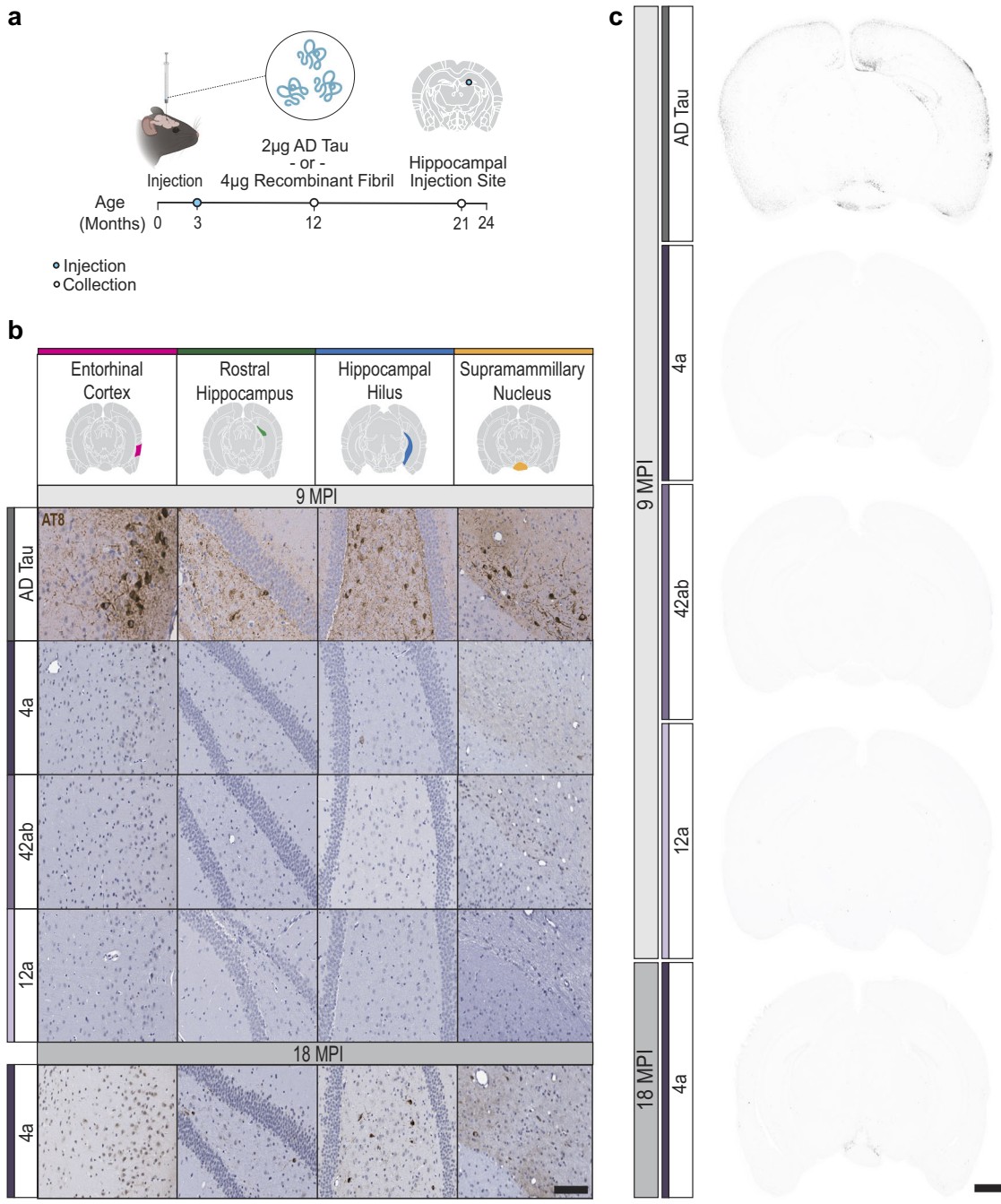

**Fig. 3 | AD PHF core alone does not replicate seeding capacity in vivo.**
**a** Experimental schematic. Wildtype mice were injected in the rostral hippocampus with AD tau or recombinant tau filaments. Mice were then aged to 9- or 18-months post injection (MPI). Schematic created in BioRender. Kasen, A. (https://BioRender. com/h8m20ur). **b** Representative tau pathology staining across 4 different brain regions. Scale bar = 100 μm. N = 8 mice per injectate at 9MPI and 3 mice at 18 MPI. **c** Representative images of pixels positive for AT8, shown in black, across the different injectates. Scale bar = 1 mm.

level of seeding induced by pronase- or phosphatase-treated tau was comparable to that of 4a recombinant tau filaments (Fig. 2), which also lack a fuzzy coat.

## The AD tau fuzzy coat is critical for seeding in vivo

We next sought to understand if cleaved or dephosphorylated AD tau showed a similar slowing of seeding kinetics to 4a tau in vivo. To test this, *MAPT* KI mice were injected with equimolar amounts of AD tau, pronase-treated AD tau, or phosphatase-treated AD tau and aged to 9 MPI (Fig. 6a). Mice brains were collected and stained for hyperphosphorylated tau pathology. Modified tau fibrils produced pathology in

the hippocampus, entorhinal cortex, and supramammillary nucleus, but total pathology was notably lower than for AD tau (Fig. 6b). Quantification of regional tau pathology throughout the midbrain showed that pathology induced by pronase- or phosphatase-treated tau was reduced up to several orders of magnitude depending on the region (Fig. 6c, d). The same brain regions were impacted by tau pathology in all three conditions, but to lesser degrees after pronase or phosphatase treatment (Fig. 6e). We found a good correlation between regional pathology seeded by 4a recombinant tau and pronase-treated tau (Fig. 6f), suggesting the tau fuzzy coat is necessary to accelerate in vivo tau seeding.

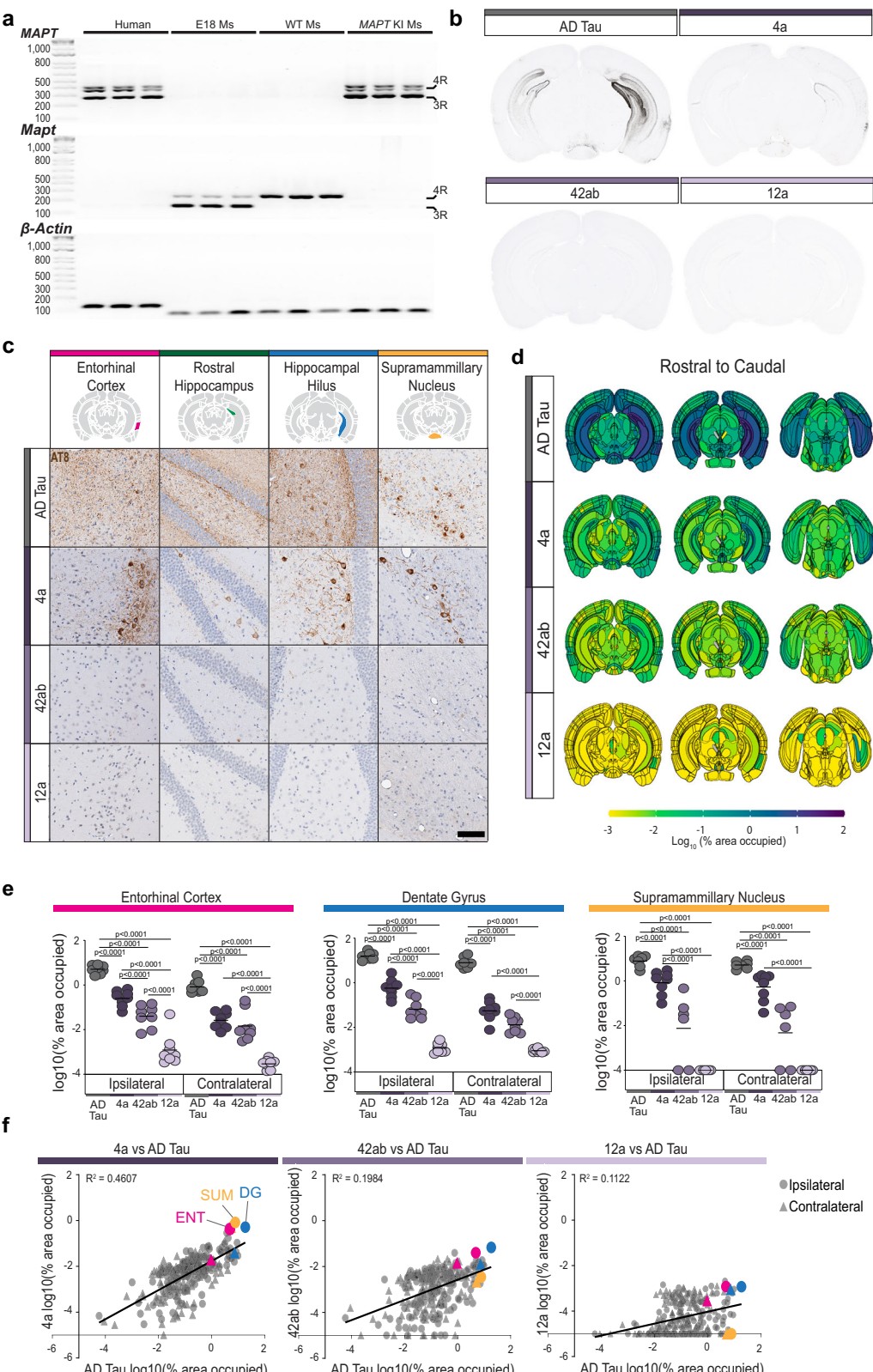

## Re-phosphorylation of AD tau rescues its seeding capacity

To further understand the role of fuzzy coat phosphorylation in tau seeding, we aimed to restore phosphorylation of dephosphorylated AD tau filaments. Several tau kinases have previously been described[44]. We chose three kinases—GSK3β, SAPK4, and CK1δ—that have been reported to phosphorylate tau at sites that become hyperphosphorylated in AD[45–48]. To determine the phosphorylation activity of

the three tau kinases, recombinant 4R tau monomer was phosphorylated in vitro[47–49]. Phosphorylated tau was evaluated by high-resolution LC-MS/MS to identify phosphorylation sites (Supplementary Fig. 6, and Supplementary Fig. 7) or transferred to membranes for western blotting (Fig. 7a). CK1δ showed minimal phosphorylation of tau (Supplementary Fig. 6). GSK3β phosphorylated tau at S396/S404 but failed to phosphorylate S202/T205. SAPK4 phosphorylated both S202/

**Fig. 4 | Mice with 3R and 4R tau show increased seeding capacity. a** The PCR products shown in Fig. 1a expanded to show the PCR products of *MAPT* KI mice. These mice do not express mouse tau but express similar human tau isoforms as the human brain. N = 3 independent brain extracts. Units in BPs (**b**) Representative images of pixels positive for AT8, shown in black, across the different injectates. **c** Representative images of tau pathology across 4 brain regions of *MAPT* KI mice injected with AD tau or recombinant tau filaments and aged to 9 MPI. Scale bar = 100 μm. **d** Pathology heatmaps representing the average pathology in each anatomical brain region at 9 MPI from the corresponding injectate. N = 8 mice/group.

**e** Quantification of pathology in the indicated brain region. Colors correspond to injectate. N = 8 mice/group. ****$p$ < 0.0001, second-generation p-values, linear regression. **f** Correlation of pathology induced by recombinant tau filaments vs AD tau. Each region is represented by a single point. The line of best fit is shown. Circles represent the ipsilateral region, and triangles represent the contralateral region. Brain regions highlighted in 4B are represented by the corresponding color: pink−entorhinal cortex (ENT), blue - dentate gyrus (DG), yellow−supramammillary nucleus (SUM). Source data are provided as a Source Data file.

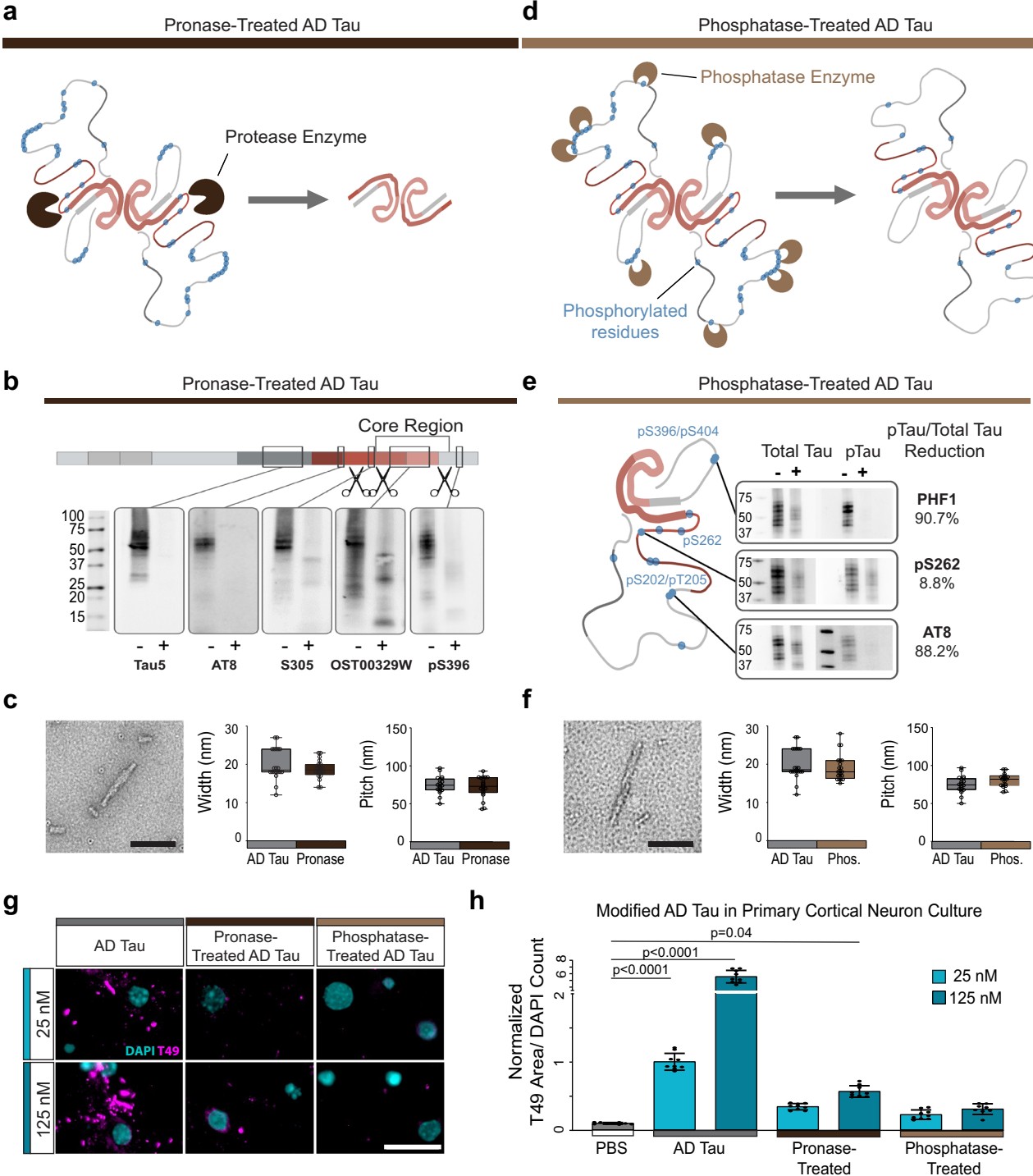

**Fig. 5 | Loss of phosphorylation or fuzzy coat reduces tau seeding capacity.**
**a** Schematic of proposed resulting filament after pronase treatment. Blue dots on the schematic are representative of potential phosphorylation sites on the AD tau filament. Schematic created in BioRender. Kasen, A. (https://BioRender.com/h8m20ur). **b** Western blot analysis of pronase-treatment of AD tau. Blots show non-treated (-) and pronase-treated (+) AD tau for 5 tau antibodies targeting various regions of the tau protein. Antibody epitopes are shown on the schematic tau monomer with scissors representing sites targeted by the pronase enzyme. N = 1 per treatment. Units in kDa. Schematic created in BioRender. Kasen, A. (https://BioRender.com/h8m20ur). **c** Representative negative stain transmission electron microscopy (TEM) of pronase-treated AD tau. Quantification of measured width and pitch of AD tau and pronase-treated AD tau. Individual filaments are represented by a single point N = 20 filaments. The five number summary of AD tau and pronase-treated AD tau respectively: minimum= 12, 14; Q1 = 18, 17.25; median= 18.5, 18.5; Q2 = 24, 20, maximum= 27, 23. **d** Schematic of proposed resulting filament after phosphatase treatment. Blue dots on the schematic are representative of potential phosphorylation sites on the AD tau filament. Schematic created in BioRender. Kasen, A. (https://BioRender.com/h8m20ur). **e** Western blot analysis of phosphatase-treated AD tau. Blots show the non-treated (-) and phosphatase-treated (+) AD tau. Antibody epitopes are highlighted on a single tau filament of the AD PHF. Tau dephosphorylation was quantified as a percentage of the initial phosphorylation level at each site. N = 1 per treatment. Units in kDa. Schematic created in BioRender. Kasen, A. (https://BioRender.com/h8m20ur). **f** Representative TEM image of the phosphatase-treated AD tau. Quantification of filament width and pitch of the AD tau and phosphatase-treated AD tau filaments. Each point represents an individual filament. N = 20 filaments. The five number summary of AD tau and phosphatase-treated AD tau respectively: minimum= 12, 15; Q1 = 18, 16.25; median= 18.5, 18; Q2 = 24, 21, maximum= 27, 28. **g** Representative images of tau seeding across doses and filaments. Scale bar = 20 μm. **h** Tau pathology, measured by T49, quantified by T49 signal relative to DAPI count. Data is presented as mean ± SEM with individual values plotted. N = 9 independent wells from 3 separate cultures. *$p < 0.05$, ****$p < 0.0001$, One-way Welch ANOVA test and Dunnett's T3 multiple comparison test compared to PBS control. Source data are provided as a Source Data file.

T205 and S396/S404 (Fig. 7a, Supplementary Fig. 6). SAPK4 does not hit the full complement of AD tau PTMs (Fig. 7b), but due to its strong phosphorylation of known pathogenic residues, SAPK4 was chosen to re-phosphorylate phosphatase-treated AD tau.

Dephosphorylated AD tau was incubated with SAPK4 using the same protocol as used for recombinant tau. Re-phosphorylation of AD tau was confirmed through western blotting for AT8 and PHF1. SAPK4 treatment restored 90% of S396/S404, but not all phosphorylation at S202/T205 and S396/S404 (Fig. 7c). Western blotting with the core tau antibody was used to quantify tau concentration following re-phosphorylation. Primary cortical neurons were treated with equimolar amounts of AD tau, dephosphorylated AD tau, or re-phosphorylated AD tau (Fig. 7d). Partial re-phosphorylation of AD tau at S202/T205 and S396/S404 rescued seeding, albeit not completely to the level of AD tau, which is consistent with the lack of full re-phosphorylation by SAPK4 (Fig. 7e).

### Recombinant PAD12 tau filaments recapitulate the seeding capacity of AD tau

A recent study showed that introducing 12 phosphomimetic mutations (Supplementary Fig. 8) in serine and threonine residues in recombinant full-length tau (PAD12 tau) enables it to fold into filaments with an ordered core that is identical to AD PHFs, and that also have a fuzzy coat[50]. We confirmed by cryo-EM that our preparation of PAD12 filaments had a core structure that was identical to that of AD tau (Fig. 8b), and used these filaments to seed primary cortical neurons, in parallel with seeding experiments with equimolar concentrations of AD tau. PAD12 filaments matched the seeding capacity of AD tau at a dose of 25 nM and, while the seeding capacity was slightly lower at a dose of 125 nM, PAD12 filaments exceeded the seeding capacity of all other screened recombinant tau filaments by at least 10-fold (Fig. 8c,d). We then sought to characterize the seeding capacity of the PAD12 in-vivo over a short period of time in wildtype mice. To test this, wildtype mice were injected with either 2 μg of AD tau or PAD12 filaments and aged to 3 MPI (Fig. 8e). The brains of the mice were collected and stained for hyperphosphorylated tau as a measure of pathology. Incredibly, the PAD12 filaments induced pathology in wildtype mice at 3MPI in key regions of the dentate gyrus, entorhinal cortex, and supramammillary nucleus (Fig. 8f), although the induced pathology was not to the same extent as AD tau (Fig. 8g, h). Similar brain regions were found to have pathology between the PAD12 and AD tau treatment, showing a conserved pattern of spread between the filaments (Fig. 8i).

While the PAD12 filaments induced cell-body like inclusions, there was notably less neuritic pathology in the outer regions of the CA1 field of the hippocampus (Fig. 8f), which was quantified as lower levels of pathology induced by PAD12 (Fig. 8h). Therefore, we wanted to assess if the lower levels of pathology in the PAD12 group were due to the lack

of neuritic pathology. We used a cell detection method in QuPath to detect all cell nuclei and then manually annotated cells as positive or negative for a cell body inclusion (Fig. 8j). The resulting cell maps were then processed through the QUINT workflow to measure differences in cell body counts. The difference in cell body like inclusions remained significantly lower in the dentate gyrus on both the ipsilateral and contralateral side; however, only the entorhinal cortex on the ipsilateral side remained significantly different and the supramammillary nucleus was not significantly different in the number of cell body like inclusions (Fig. 8k). Only one other publication has shown seeding in wildtype mice after 3 MPI; however, this group injected 4.5 times the amount of recombinant protein than what was injected with PAD12 in this study[24]. This suggests that the seeding capacity of the PAD12 filaments is the closest to AD tau than other recombinant fibrils, making it a useful tool in understanding tau seeding pathology in AD.

## Discussion

Seeded aggregation is thought to underline the progression of tau pathology throughout the brain in AD, yet the factors that determine the efficiency of this process remain poorly understood. Tau filament structure varies between tauopathies, but recombinant tau filaments induced by co-factors form structures distinct from known disease-associated structures. These recombinant filaments fail to generate AD-like pathology in wildtype mice, highlighting the importance of filament structure in tau seeding. In the current study, we sought to understand barriers to the seeding capacity by addressing two questions: How does filament core structure impact seeding? And do the disordered regions of tau surrounding the structured filament core play a role in this process?

We first explored the role of the core structure on seeding efficiency. We found that recombinant tau filaments replicating the core structure of AD PHF show enhanced seeding capacity compared to filaments with different core structures. This finding aligns with previously published data from biosensor cells where the use of alanine codon substitutions in tau filaments revealed seeding differences between structurally distinct filaments, with core structures similar to AD PHF showing a seeding pattern similar to AD tau[51]. Despite replicating the correct core structure, 4a recombinant filaments showed lower seeding capacity than AD tau in both the primary cortical neurons and mice. Together, these observations suggest that the core structure of tau filaments plays an important, but not exclusive role in seeding capacity.

We then explored the role of the fuzzy coat on seeding capacity. By modifying AD tau with pronase or phosphatase, we showed that the phosphorylation of the fuzzy coat also plays a role in tau seeding capacity. To confirm this finding, we used a kinase to re-introduce phosphorylation sites on phosphatase-treated AD tau and found a

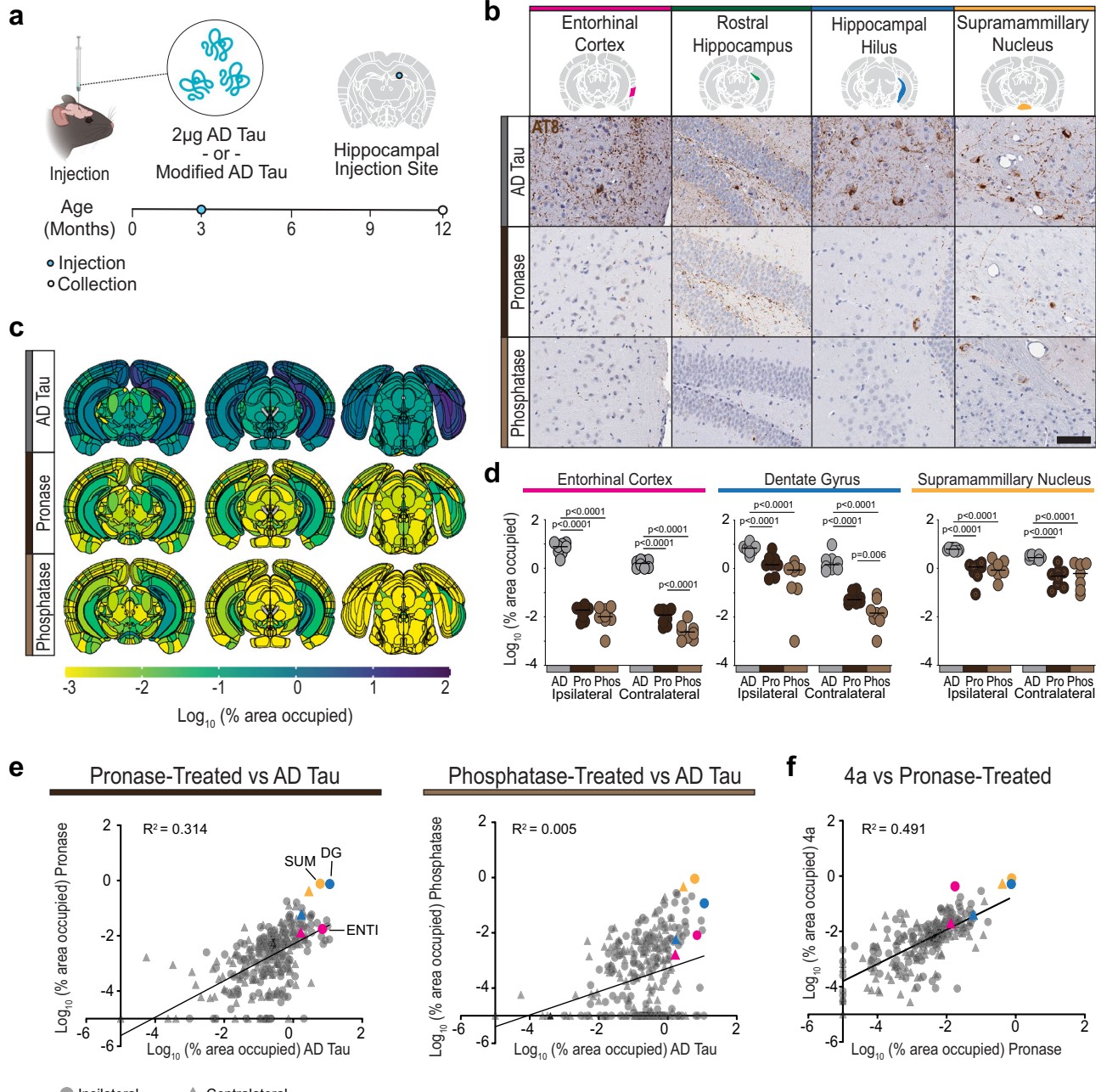

**Fig. 6 | AD tau fuzzy coat is critical for seeding in vivo. a** Experimental schematic. *MAPT* KI mice were injected with AD tau or modified AD tau filaments in the rostral dentate gyrus. Mice were then aged 9 MPI. Schematic created in BioRender. Kasen, A. (https://BioRender.com/h8m20ur). **b** Representative images of pathology, measured by AT8, across 4 brain regions. Scale bar = 100 µm. **c** Pathology heatmaps representing the average pathology in each anatomical brain region at 9 MPI from the corresponding injectate. *N* = 8 mice per injectate. **d** Quantification of pathology in the indicated brain region (see "Methods"). Colors correspond to injectate. *N* = 8 mice per injectate *\*p* < 0.05, \*\*\*\**p* < 0.0001, second-generation p values based on a null interval of ± 5% difference with 95% confidence intervals. **e** Correlation of

pathology induced by modified AD tau filaments vs AD tau. Each region is represented by a single point. Circles represent the ipsilateral region, and triangles represent the contralateral region. Brain regions highlighted in 6D are represented by the corresponding color: pink−entorhinal cortex (ENT), blue - dentate gyrus (DG), yellow−supramammillary nucleus (SUM). Source data are provided as a Source Data file. **f** Correlation of 4a filaments to pronase-treated AD tau. Each region is represented by a single point. Circles represent the ipsilateral region, and triangles represent the contralateral region. Brain regions highlighted in 6D are represented by the corresponding color: pink−entorhinal cortex (ENT), blue - dentate gyrus (DG), yellow−supramammillary nucleus (SUM).

partial rescue in seeding capacity. Furthermore, we showed that full-length recombinant tau filaments that recapitulate the AD tau core structure *and* disease-associated PTMs with phosphomimetic mutations in the fuzzy coat recapitulated the seeding efficiency of AD tau.

How do PTMs in the fuzzy coat confer increased seeding capacity? The tau monomer, in its native state, is intrinsically disordered and

does not form filaments efficiently due to its propensity to adopt a 'paperclip' conformation, which prevents aggregation[52–55]. Over the course of AD progression, tau accumulates many PTMs, predominantly in the N- and C-termini, with phosphorylation being the most prevalent[9,41,56,57]. Interestingly, when phosphorylation sites or phosphomimetic mutations are added to the tau monomer, the full-

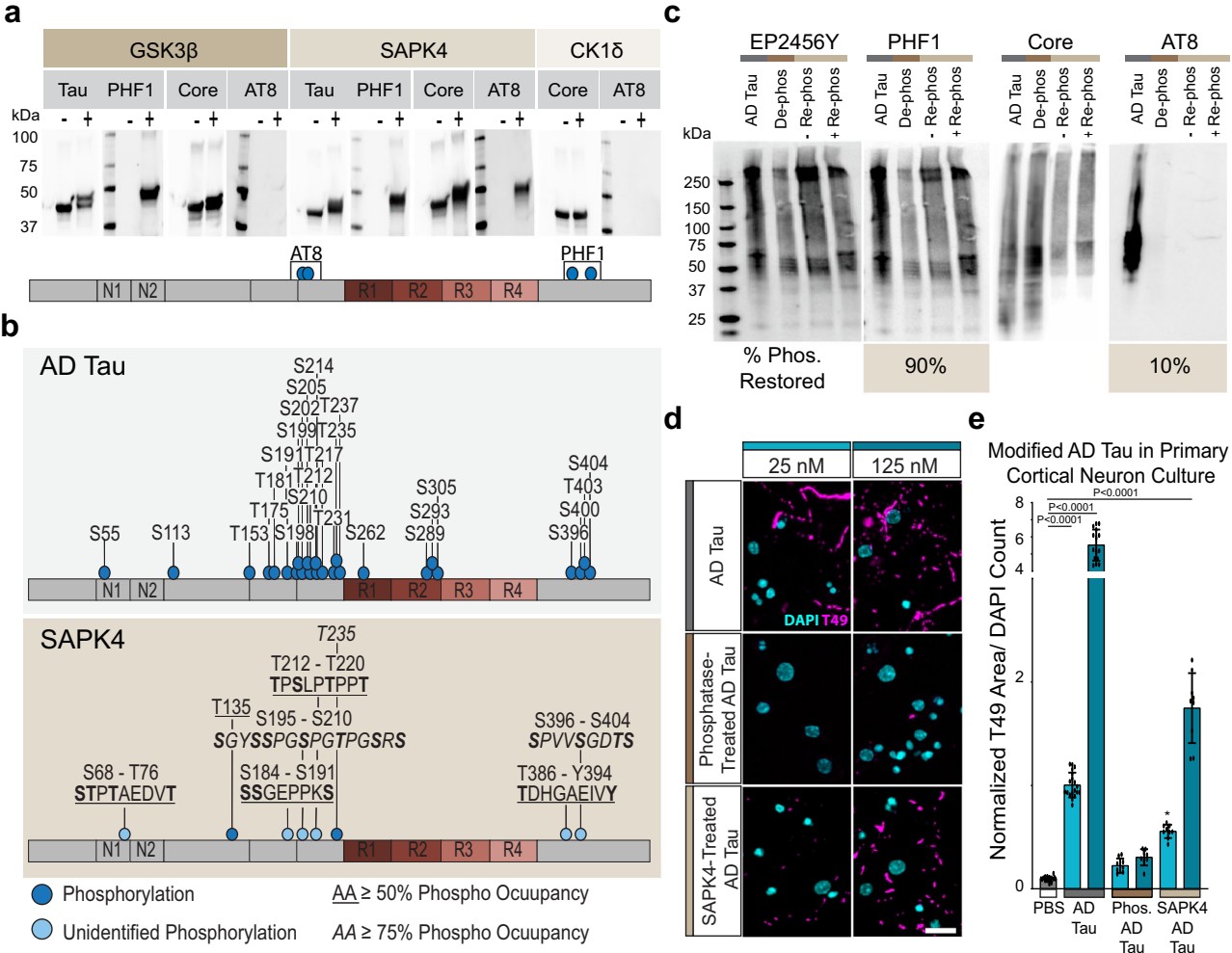

**Fig. 7 | Re-phosphorylation of AD tau rescues its seeding capacity. a** Western blots of tau monomer treated with GSK3β, SAPK4, or CK1δ for phosphorylation of S202/T205 and S396/S404. Each blot shows the non-treated (-) and treated (+) tau monomer. N = 1 per treatment. **b** Schematic of phosphorylation sites on AD PHFs, previously published[9], and phosphorylation of tau monomer by SAPK4. Single phosphorylation sites are marked by a dark blue dot and an unidentified phosphorylation site is indicated by a light blue. For each unidentified phosphorylation site, the sequence is noted with the possible phosphorylated amino acid in bold. Phospho-occupancy was measured in from a non-quantitative LC-MS/MS. Resulting peptide-spectrum matches (PSMs) were aligned to the tau sequence. At each amino acid, phospho-occupancy was estimated from the ratio of modified to total detected peptides[86]. Amino acids with greater than 50% phospho-occupancy are underlined. Amino acids with greater than 75% phosphor-occupancy are italicized.

**c** Western blots of AD tau, phosphatase-treated AD tau (De-phos), non-treated phosphatase treated AD tau (- Re-phos), and SAPK4-treated phosphatase-treated AD tau (+Re-phos) for phosphorylation of S202/T205 and S396/S404. The percent phosphorylation restored is calculated by the percent difference in the ratio of pTau to total tau signal of the + Re-phos lane compared to the AD tau. N = 1 per treatment. **d** Representative images of primary neuron cultures treated with AD tau or modified tau filaments at 2 doses and maintained for 21 days after treatment. Scale bar = 20 μm. **e** Tau pathology quantified as T49 signal relative to DAPI count. Data is presented as mean ± SEM with individual values plotted. N = 9 independent wells from 3 separate cultures. *$p < 0.05$ ****$p < 0.0001$, One-way Welch ANOVA test and Dunnett's T3 multiple comparison test comparing to PBS control. Source data are provided as a Source Data file.

length protein will form filaments[45,50,58–60]. The phosphorylation in the fuzzy coat is thought to destabilize the 'paperclip' fold through changes in the charge distribution of the protein, thereby promoting its assembly into amyloid filaments[52–55,61]. The question remains: how does tau phosphorylation facilitate seeded aggregation?

Seeding, in a biological context, consists of multiple processes: 1) internalization, 2) trafficking through the endolysosomal system, 3) escape from the lysosome, and 4) templated assembly of monomers into filaments within the cytosol. While our assays do not distinguish between these processes, we discuss the potential relevance of fuzzy coat PTMs to each of these processes. Previous studies have found that tau can be internalized following cell surface interactions with heparin sulfate proteoglycans[62–64] or other more specific receptors such as LRP1, integrin αvβ1, and EGFR3[65–68]. The negative charge shift induced by phosphorylation could facilitate interaction with specific proteins,

although both HSPGs and the plasma membrane have a negative charge which would be expected to repel the more negative tau filaments. In addition, cross-filament interactions of tau could generate larger tau seeds that are not able to be internalized due to size constraints. Previous work has noted that the recombinant filaments made from the truncated tau construct (297-391), such as 4a and 12a, tend to clump together over time, while PAD12 filaments do not[50]. Our filament measurements show some differences, although all filaments had a majority population within the size window expected to be internalized by cells[69]. Once inside the cell, phosphorylation of the filament may aid recruitment by providing a negative net charge that attracts positively charged monomers, facilitating their incorporation into the filament.

One complication of this study was the need to compare truncated fibrils to full-length tau fibrils, including AD tau lysate, which also

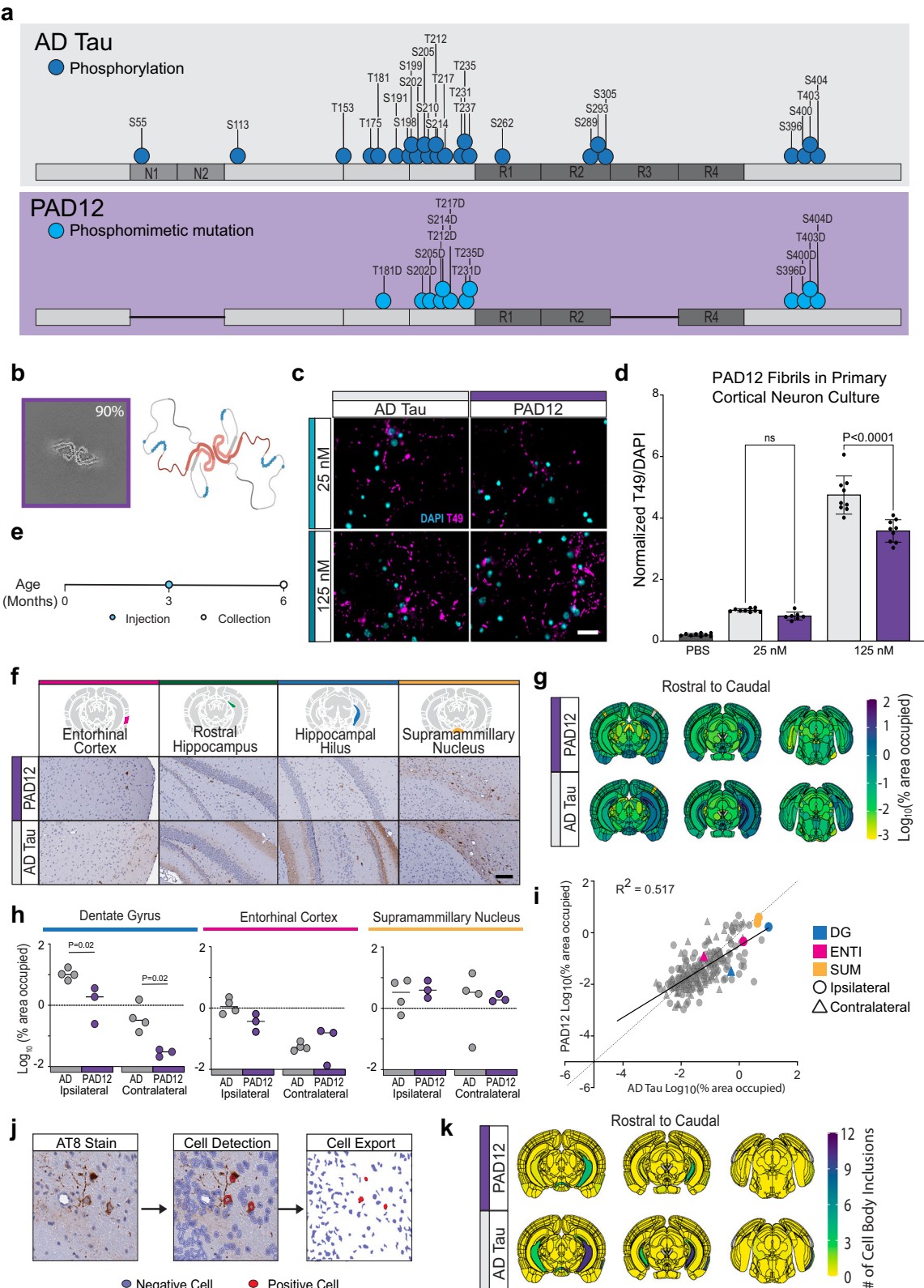

contains additional proteins. We resolved the size discrepancy by treating neurons with equimolar levels of tau, with the rationale that matching the tau number should provide equivalent opportunities for seeding. Our study used well-established metrics of pathogenic tau (detergent insolubility, hyperphosphorylation) to measure pathology, but we do not show that the seeded tau pathology has the same structure as the seeds themselves. While we hypothesize that fibrils

seed pathology with a similar core structure, testing this in future studies will provide additional information about the fidelity of seeding in animal models.

In summary, our data indicate that both the structure of the filament core and post-translational modifications in its fuzzy coat alter tau's seeding capacity. Several open questions remain. The molecular mechanisms by which the AD core structure and phosphorylation of

**Fig. 8 | Recombinant tau with the PHF core and 12 phosphomimetic sites recapitulates the seeding capacity of AD PHF. a** Schematic of the comparison of phosphorylation sites on AD tau[9] and PAD12 filaments[50] with a phosphorylation site represented by a dark blue circle and the phosphomimetic mutation represented by a light blue circle. **b** Cryo-EM average of PAD12 filaments. Schematic created in BioRender. Kasen, A. (https://BioRender.com/h8m20ur). **c** Representative images of primary cortical neuron culture treated with sonicated AD tau or PAD12 filaments and maintained for 21 days post treatment. Scale = 20 μm. **d** Tau pathology quantified by T49 area relative to DAPI count. Data is presented as mean ± SEM with individual values plotted. $N = 9$ independent wells from 3 separate cultures, $p$ values represent fold-change to the corresponding AD tau dose with One-way Welch ANOVA test and Šidák's multiple comparison test: ****$p < 0.0001$. **e** Schematic of injection timeline. Wildtype mice were injected with 2 μg of AD tau or PAD12 filaments in the rostral dentate gyrus and aged to 3 MPI. **f** Representative images of pathology, measured by AT8, across 4 brain regions. Scale bar = 100 μm.

**g** Pathology heatmaps representing the average pathology in each anatomical brain region at 3 MPI from the corresponding injectate (see "Methods"). $N = 4$ mice AD tau $N = 3$ mice PAD12. **h** Quantification of pathology in the indicated brain region. Colors correspond to injectate. *$p < 0.05$, second-generation p values based on a null interval of ± 5% difference with 95% confidence intervals. **i** Correlation of pathology induced by PAD12 filaments vs AD tau. Each region is represented by a single point. Circles represent the ipsilateral region, and triangles represent the contralateral region. Brain regions highlighted in 8 F are represented by the corresponding color: pink– entorhinal cortex (ENT), blue–dentate gyrus (DG), yellow–supramammillary nucleus (SUM). **j** Representative images depicting cell nuclei detection on stained tissue to classify cells as positive (red) or negative (purple) for AT8 cell-body-like pathology. **k** Pathology heatmaps representing the average number of cells positive cell-body-like inclusions in each anatomical brain region at 3 MPI from the corresponding injectate. $N = 4$ mice AD tau $N = 3$ mice PAD12. Source data are provided as a Source Data file.

the fuzzy coat impact the seeding capacity of tau have yet to be elucidated, and future work will be necessary to determine if phosphorylation in the fuzzy coat also affects seeding in non-AD tauopathies. Meanwhile, recombinant PAD12 filaments provide a convenient tool to induce tau seeding in model systems, which will improve our understanding of progressive tau pathology in the context of Alzheimer's disease. Our results also suggest that dephosphorylation of tau filaments may represent a disease-modifying strategy to slow tau pathology progression, although specific targeting of tau filaments would require further development[70–72].

## Methods

### Animals
C57BL/6J mice were purchased from the Jackson Laboratory (000664; RRID:IMSR_JAX:000664). CD-1 mice used for primary neuron culture were purchased from Charles River (CD1(ICR); RRID:IMSR_CRL:022). *MAPT* KI mice (B6.Cg-Mapt^tm1.1(MAPT)Tcs) mice were provided by the RIKEN BRC through the National BioResource Project of the MEXT Japan (RBRC09995; RRID:IMSR_RBRC09995). All mice were housed at 22 °C and 50% humidity with 12-h light cycles (lights on at 7am, lights off at 7 pm). Male and female mice were used and were 3–4 months old at the time of injection. Mice were aged 12–21 months old prior to sample collection. 24 female and 24 male C57BL/6 J mice and 24 female and 24 male *MAPT* KI mice were utilized in the filament injection studies.

### Alzheimer's disease tau extraction and characterization
All procedures were completed in accordance with local institutional review board guidelines of Van Andel Institute. AD brains with high frontal cortical burden of tau pathology were identified from the VAI Brain Bank (RRID:SCR_026035) by immunohistochemical staining. After extracting all cases and confirming the seeding capacity of tau, cases were pooled for use in subsequent experiments.

Tau was extracted from selected AD brains following an established protocol[24,73]. After thawing cortical tissue, meninges were removed, and the gray matter was separated from the white matter. The gray matter was weighed and suspended in nine volumes (w/v) of high-salt buffer (10 mM Tris-HCL (pH 7.4), 800 mM NaCl, 1 mM EDTA, 2 mM dithiothreitol (DTT), PMSF, and protease and phosphatase inhibitors) with 0.1% sarkosyl and 10% sucrose. The tissue was homogenized with a dounce homogenizer and centrifuged at $10,000 \times g$ for 10 min at 4 °C. The pellet was resuspended with the same buffer conditions and the supernatants from all extractions were filtered and pooled together.

Additional sarkosyl was added to the pooled supernatant to a final concentration of 1% and the supernatant was stirred for 1 h at room temperature. The sample was centrifuged at $125,000 \times g$ for 75 min at 4 °C. The resulting pellet, containing pathological tau, was washed

three times with PBS and resuspended in 39 mL of PBS. The pellet was centrifuged at $180,000 \times g$ for 30 min at 4 °C. The supernatant was removed, and the resulting pellet was resuspended in 100 μL DPBS per 1 gram of gray matter. The suspension was briefly vortexed and centrifuged on a tabletop centrifuge at $1000 \times g$ for 1 min. The resulting sample was rocked overnight at 4 °C on the Thermo Fisher Labquake rotator. The sample was then centrifuged at $1000 \times g$ for 1 min at 4 °C on a tabletop centrifuge.

The resuspended pellet was passed through a 27 G/0.5-inch needle to resuspend the pellet. The solution was transferred to an autoclaved 1.5 mL Beckman ultracentrifuge tube (Beckman Colter REF: 357448) and an additional 100 μL of DPBS was added to the sample. The sample was centrifuged in the OptimaMAX-TL centrifuge, in the TLA100.3 rotor, at $80,000 \times g$ for 30 min at 4 °C. The supernatant was removed and 100 μL DPBS/ 1 g of gray matter was added to the pellet. The sample was vortexed and further resuspended by a brief sonication (QSonica Q55; CL-188 probe; 20 pulses; amplitude 25; 1 second pulse). The sample was centrifuged in the OptimaMAX-TL centrifuge, in the TLA100.3 rotor, at $80,000 \times g$ for 30 min at 4 °C. The pellet was resuspended with 20% of the removed volume of DPBS. The pellet was resuspended by sonication (QSonica Q55; CL-188 probe; 20 pulses; amplitude 25; 1 second pulse) and centrifuged at $10,000 \times g$ for 30 min at 4 °C. The final supernatant was utilized for all studies and is referred to as AD tau. The extraction was characterized by western blotting for tau (described below). For the extraction used in this study, tau constituted 20.9% of the total protein.

### Recombinant tau protein expression and purification
In this study the following tau constructs were used; 297-391, 297-408 S396D S400D T403D S404D and 0N3R T181D S202D T205D T212D S214D T217D T231D S235D S396D S400D T403D S404D (phospho-mimetic of AD 12, or PAD12 tau). The expression and purification of recombinant tau has been described. Tau plasmids were transformed into BL21 gold E. *coli* cells (Agilent, Cat number 200131) and grown in 2xTY (tryptone yeast) supplemented with 100 mg/mL of ampicillin to an optical density of 0.8. Subsequently the cells were induced for three h at 37 °C by the addition of 0.8 mM IPTG. Cells were pelleted by centrifugation at $4400 \times g$ for 30 min at 4 °C and the pellets were snap-frozen using liquid nitrogen, placed in a beaker and resuspended in 50 mM MES, pH 6.0, 50 mM NaCl, 5 mM DTT, 5 mM EDTA, supplemented with 0.1 mM phenylmethylsulphonyl fluoride, cOmplete EDTA-free protease cocktail inhibitors, 40 μg/mL DNAse I and 10 μg/mL RNAse, at 10 mL per gram of pellet and heated to 95 °C for 5 min. Cells were lysed by sonication using a metal sonicator probe (Sonics VCX-750 Vibracell Ultra Sonic Processor) at an amplitude of 40% 5 seconds on, 10 seconds off for 5 min followed by continuous 30 s sonication at 90 % amplitude. Lysed cells were centrifuged at $20,000 \times g$ for 30 min at 4 °C. The supernatant was loaded onto a 5 mL

HiTrap-CaptoS column (GE Healthcare) Äkta Start and eluted using 1 M NaCl. Fractions were analyzed by SDS-PAGE and protein containing fractions were pooled and precipitated using 0.38 g of ammonium sulfate per mL and left on a rocker for 30 min at 4 °C. The precipitated protein was centrifuged at 20,000 × $g$ for 30 min at 4 °C. The supernatant was discarded, and the protein was resuspended in 2 mL of 10 mM KPOH, pH 7.2, 10 mM DTT and loaded onto a 16/600 75-pg size-exclusion column. Fractions were analyzed by SDS-PAGE and protein containing fractions were pooled and concentrated to 5–10 mg/mL using molecular weight concentrators with a cutoff filter of 3 kDa. Purified protein samples were flash frozen in liquid nitrogen. Protein concentrations were determined using NanoDrop2000 (Thermo Fisher Scientific).

### Recombinant tau filament assembly

Filaments were assembled as previously described[25,50]. Assembly reactions were carried out in 40 µL aliquots in a 384-well microplate that was sealed and placed in a Fluostar Omega (BMG Labtech). Tau297-391 filaments were assembled in 10 mM PB pH 7.2 10 mM DTT with either 100 mM $MgCl_2$ or 100 µM $CuCl_2$ using a 200 rpm orbital shaking for 24 h. Tau297-408 S396D S400D T403D S404D were assembled in 10 mM PB pH 7.2 10 mM DTT 100 mM $MgCl2$ using 200 rpm orbital shaking for 48 h. 0N3R PAD12 were assembled in 40 mM HEPES, pH 7.28, 250 mM citrate, 4 mM TCEP using 500 rpm 1 min on 1 min off orbital shaking for 76 h.

### Pronase treatment of AD tau

The AD tau sample was diluted to 0.125 mg/mL tau in DPBS. Pronase (EMD Millipore, #537088) was added to a final concentration of 0.4 mg/mL. The sample was incubated at 37 °C for 2.5 min. The reaction was stopped by adding a 2x solution of c0mplete protease inhibitor cocktail (Sigma Aldrich, REF:11836170001) and briefly vortexed. The sample was centrifuged at 80,000 x $g$ for 30 min in the Optima Max Centrifuge at 4 °C. The supernatant, containing pronase, was removed, and the pellet, containing tau, was resuspended in 300 µL PBS and centrifuged at 80,000 × $g$ for 30 min in the Optima Max Centrifuge at 4 °C. This was repeated for a total of 3 centrifugation steps. The final pellet was resuspended in 25% of the initial reaction volume. The protein concentration of each saved fraction was calculated with the Pierce BCA Protein Assay Kit (Thermo Fisher, #23227). Fractions were analyzed by total protein gel and western blotting to track pronase and purified tau.

### Alkaline phosphatase treatment of AD tau

The AD tau sample was diluted to a total protein concentration of 1 mg/mL. The FastAP Thermosensitive Alkaline Phosphatase kit (Thermo Fisher, #EF0651) was used, adding the alkaline phosphatase enzyme at 1 U/µg AD tau. C0mplete protease inhibitor cocktail (Sigma Aldrich, #11836170001) was added to the reaction to prevent protein degradation. The reaction was run at 32 °C for 18 h with shaking at 500 rpm. The reaction was stopped by the addition of EDTA. The sample was centrifuged at 80,000 x $g$ for 30 min in the Optima Max Centrifuge at 4 °C. The supernatant, containing the phosphatase, was removed, and the pellet, containing tau, was resuspended in 300 µL PBS and centrifuged at 80,000 × $g$ for 30 min in the Optima Max Centrifuge at 4 °C. This was repeated for a total of 3 centrifugation steps. The final pellet was resuspended in 25% of the initial reaction volume. The protein concentration of each saved fraction was calculated with the Pierce BCA Protein Assay Kit (Thermo Fisher, #23227). Fractions were analyzed by total protein gel and western blotting to track phosphatase and purified tau.

### Re-phosphorylation of phosphatase-treated AD-tau

Phosphatase-treated AD tau or recombinant tau monomer was diluted to a final concentration of 0.33 mg/mL in re-phosphorylation buffer (50 mM Tris-HCl (Invitrogen, REF:15567), 10 mM $MgCl_2$ (Fisher, BP214-500), 5 mM Tris-buffered ATP (Thermo Scientific, Cat#:R1441), 1 mM EDTA (Invitrogen, Cat#:15575-020), 1 mM DTT (BioRad, Cat#1610610), 1 mM sodium orthovanadate (Fisher, S454-50), 0.2 mM PMSF (Millipore Sigma, Cat#: 329-98-6), 1 µg/mL pepstatin (Roche, REF:11359053001), and c0mplete protease inhibitor (1 tablet/10 mL) (Roche, #11873580001)). The enzyme was added to the reaction [0.4 U per 30 µL SAPK4 (Millipore; Item 14-249; Lot# D8EN044U-C), 0.1 U per 30 µL GSK3β (SignalChem; G09-10G; Lot# E3081-10), or 300 mM CK1δ(SignalChem; C65-10G; Lot# J4063-7)]. The reaction was shaken at 400 RPM at 30 °C for 2.5 h (Eppendorf ThermoMixer F1.5). At the end of the reaction, samples were centrifuged at 80,000 × $g$ at 4 °C for 30 min and the pellet was resuspended in DPBS.

### Western blot

All samples were prepared by diluting in 2x Laemmli Sample Buffer (Bio Rad, #1610737) with 5% β-mercaptoethanol (Sigma-Aldrich, #M3148) and boiling for 10 min at 95 °C. Samples were loaded at 20 µg of protein per lane in 7.5% or 4–20% Mini-PROTEAN TGX Precast Gel (Bio Rad; #4561026, #4561096) and protein gels were run at consistent voltage of 0.02 mA per gel for 50 min. Gels were washed in $diH_2O$ for 1 min and then washed in cold transfer buffer with a nitrocellulose membrane for 15 min. Proteins were then transferred to the nitrocellulose membrane at 90 V for 90 min at 4 °C. Membranes were then washed in sterile water for 3 min and total protein was visualized with ponceau stain and imaged on the Bio Rad Chemidoc Imaging System (BioRad, #12003153). The membrane was then washed in TBS with 0.1% Tween (TBST) for 3 × 5 min and blocked in 5% non-fat dry milk (NFDM) in TBST for 1 h at room temperature. The membrane was washed 3 × 5 min in TBST and primary antibodies (see Supplementary Table 1), diluted in 5% NFDM, were incubated overnight at 4 °C. Membranes were then washed 3 × 5 min in TBST and secondary fluorescent antibodies were incubated at room temperature for 1 h in the dark. Membranes were washed in TBST 3 times for 5 min and then imaged.

The ImageLab software was used to calculate the relative phosphorylated tau signal to total tau signal. The whole lane was outlined to detect fluorescent signal for the phosphorylated and total tau channels for each sample. The phosphorylated tau signal was divided by the total tau signal to obtain the phosphorylated to total tau ratio. These ratios were then divided by the ratio of the control sample to obtain the normalized phosphorylated tau to total tau ratio, presented in the graphs.

### Negative stain electron microscopy

Tau filaments were diluted to 0.2 mg/mL tau in PBS. Carbon grids (Electron Microscopy Sciences, #CF300-Cu) were prepared with glow discharge for 30 seconds with $O_2$ (Gatan Solarus Model 950). With the foil side of the grid facing up, 3 µL of the diluted tau was added to the grid and left for 1 min. The grid was blotted with filter paper. 20 µL of 1% uranyl acetate was added to the grid and immediately blotted with filter paper. 20 µL of sterile water was added to the grid and immediately blotted with filter paper twice. 20 µL of 1% uranyl acetate was added to the grid and left for 40 seconds before blotting with filter paper. The grid was left to dry for 5 min prior to storage and subsequent imaging with the Tecnai Spirit G2 BioTWIN Transmission Electron Microscope. Representative images were taken at 30,000x with lower power images taken at 18,000x for characterization. Pitch and width were measured from 25 filaments across 5 images from 5 different windows. For filament measurement of pre and post sonication, three images at 11,000x were acquired from 30 windows across the grid. Filaments were measured in ImageJ with the minimum length set to 15 nm, which would correspond to 32 monomers[74].

## Cryogenic electron microscopy acquisition

3 μL of the reaction mixture were applied to glow-discharged cryo-EM grids (1.2/1.3 μm, 200 or 300 mesh carbon Au grids, Quantifoil). Grids were blotted and plunge-frozen into liquid ethane cooled to −180 °C using a Vitrobot IV (Thermo Fisher Scientific). Cryo-EM data were acquired on a Krios G2 (Thermo Fisher Scientific) electron microscope, operated at an accelerating voltage of 300 kV. Images were recorded using EPU software on a Falcon4i camera without an energy filter, using a dose of 30 electrons per square ångström and a pixel size of 0.824 ångström. Images were converted from eer into tiff prior to processing using relion_convert_to_tiff.

## Cryogenic electron microscopy data processing

All data were processed in RELION[75] using version 5.0. Video frames were gain-corrected using RELION'S own implementation of motion correction. Contrast transfer parameters were estimated using CTFFIND4.0[76]. Filaments were auto-picked using a modified version of Topaz[23,77] and particles were extracted using a box size of 768 pixels downscaled to 128 pixels. Reference free 2D classification was carried out to assess filament quality and the degree of polymorphism. 3D initial models were generated de novo by selecting classes showing one full crossover of the filament and using relion_helix_inimodel2d[78]. Subsequently, filaments which showed optimal 2D classes were re-extracted using a box size of 384 binned to 192 and 3D-autorefined using the initial model generated, lowpass filtered to 10 Å. Further 3D auto-refinements searching for optimal twist and rise parameters were used to improve the resolution of the reconstructions. Figures 2 and 4 show cross-section projections of the 3D reconstructions, with an approximate thickness of 4.8 Å, orthogonal to the helical axis (with a box size of 192 pixels and a pixel size of 1.648 Å).

## Dynamic light scattering

Samples were diluted to 500 nM, sonicated (Diagenode Bioruptor Pico; 10 cycles; setting medium; 30 seconds on, 30 seconds off), then further diluted to 25 nM. DLS measurements were read at 25 °C on a Dyapro Nanostar Laser Photometer (WDPN_10) using the 6613.4 nm laser. Dynamics software was used to output percent number.

## Phospho-proteomics sample prep and instrumentation

Bands of interest (n = 1 per treatment) were excised and digested with a commercially available kit (Thermo Fisher Scientific, Cat# 89871). Briefly, the bands of interest were destained at 37 °C for 90 minutes, reduced at 60 °C for 10 minutes and alkylated at 22 °C for 60 minutes according to manufacturer's instructions. Gel pieces were washed with destaining solution and then digested overnight at 30 °C with Trypsin. The primary peptide containing fraction was transferred to an auto-sampler vial (Thermo Fisher Scientific, Cat# 6PSV9-03FIVP and 6PSC9STS1R) and a secondary extraction of the gel pieces was done with 1% formic acid (Fisher Scientific, Cat# A11710X1-AMP) by incubating at 22 °C for 5 minutes. This secondary extraction was added to the same autosampler vial. The samples were dried down in a rotary vacuum evaporator (SP Scientific) for 2 h and stored at −80 °C until analysis. Samples were resuspended with 10 μL of 0.1% trifluoroacetic acid (Fisher Scientific, Cat# LS119-500) prior to injection.

## LC-MS/MS analysis

Samples were analyzed on an Exploris 480 Mass Spectrometer (Thermo Scientific) coupled to a Vanquish Neo Nano liquid chromatography system (Thermo Scientific). Peptides were separated with a C18 nano separation column (20 cm × 75 μm ID; P/N HEB07502001718IWF, CoAnn Technologies) connected to the customized heater(Phoenix S&T). Peptides were eluted using a linear gradient from 3%B to 25%B in 20 min, 80%B in 5 min, and 95%B in 5 min, for a total gradient length of 30 min. Data-Dependent Acquisition was used with a full MS scan range of 350 to 1200 m/z, resolution of 120 K, and standard maximum injection time and auto gain control. Then, a series of MS2 scans were acquired for the most abundant ions from the MS1 scan using an orbitrap analyzer with a 2-second cycle time. Ions were selected with charges between 2–5 and quadruple isolation window was set at 1.6 m/z. Ions were fragmented using higher-energy collisional dissociation (HCD) with a collision energy of 30%.

## Phospho-proteomics data analysis

Proteome Discoverer 3.1 (Thermo Scientific) was used to process the raw spectra. The search criteria are protein database; Target protein sequence (Uniprot accession number: A0A024RA17_Human Tau), carbamidomethylated (+57 Da) at cysteine residues for fixed modifications, phosphorylation at serine, threonine, and tyrosine (+80 Da), oxidized at methionine (+16 Da) residues for variable modifications, two maximum allowed missed cleavage, 10 ppm tolerance for MS and MS/MS. The false discovery rate (FDR) was controlled at 1%.

## Tau Isoform PCR

The Qiagen RNAeasy kit was used to extract RNA from bulk cortex tissue from fetal CD1 mouse brain, bulk cortex tissue C57BL/6 J adult mouse brain, human cortex, and bulk cortex tissue MAPT KI mice. Three RNA samples from each tissue were used for PCR. The following primer sets were used for the analysis:

Mouse Tau [F:5'-CCTAAGTCACCATCAGCTAGTAAG-3' R:5'-CCAC CGGCTTGTAGACTATTT-3']

Human Tau [F:5'- AAGTCGCCGTCTTCCGCCAAG −3' R:5'- GTCG GACCCAATCTTCGA −3']

β-Actin (mouse) [F:5'- GATGCCCTGAGGCTCTTTTC-3' R:5'- GCAC TGTGTTGGCATAGAGG-3']

β-Actin (human) [F:5'-TCCACGAAACTACCTTCAACT-3' R:5'-CAGT GATCTCCTTCTGCATCC-3']

The QIAGEN OneStep RT-PCR Kit was used for PCR analysis. The PCR program for the tau samples was performed as follows: a reverse transcription step of 30 min at 50 °C, then an initial PCR activating step for 15 min at 95 °C, followed by 27 cycles of 1-min steps of denaturing at 94 °C, annealing at 57 °C, and extension at 72 °C, and finally a final extension step for 10 min at 72 °C. The PCR program for the β-actin samples was performed as follows: reverse transcription step of 30 min at 50 °C, then an initial PCR activating step for 15 min at 95 °C, followed by 27 cycles of 1-min steps of denaturing at 94 °C, annealing at 57 °C, and extension at 72 °C, and finally a final extension step for 10 min at 72 °C. 5x loading dye was added to all samples and the samples were run in a 2% agarose gel with sybersafe for 1 hour at 125 V and visualized (ChemiDoc MP Imaging System).

## Stereotaxic injections

AD tau or recombinant tau filaments were vortexed and diluted with PBS to 2 mg/mL or 4 mg/mL and sonicated in a cooled bath sonicator at 9 °C (Diagenode Bioruptor Pico; 10 cycles; setting medium; 30 seconds on, 30 seconds off). Mice were injected at 3–4 months old. Mice were deeply anesthetized with isoflurane and injected unilaterally into the right forebrain targeting the dorsal hippocampus (coordinates: −2.5 mm relative to bregma, −2.0 mm from the midline, −1.8 mm beneath the dura). Injections were performed using a 10 μL syringe (Hamilton 7635-01, NV) with a 34-gauge needle (Hamilton 207434, NV) injecting 2 μg AD tau or 4 μg recombinant tau filaments (2.5 μL) at a rate of 0.4 μL/min. After the designated incubation time, mice were perfused transcardially with 0.9% saline, followed by 4% paraformaldehyde (PFA). Brains were removed and underwent overnight fixation in 4% PFA. After fixation, tissues were processed into paraffin through sequential dehydration and perfusion with paraffin under vacuum. Processed brains were embedded in paraffin blocks, cut into 6 μm sections, and mounted on glass slides.

## Primary cortical neuron culture

Primary neuron cultures were prepared from embryonic day 18 (E18) CD1 (strain 022; RRID: IMSR_CRL:022) mice. Brains were removed from the embryos and placed into a petri dish filled with ice-cold, sterile Hibernate Medium (Cat#. A1247601, Gibco). Meninges were carefully removed, and the hemispheres were separated. The hippocampus was removed from each hemisphere and the cortex was cut into 2x2mm sections. Cortices from several brains were pooled and digested in papain solution (20 U/mL Cat# LS003126, Worthington) and then treated with DNase I (Cat# LS006563, Worthington) to remove residual DNA. The tissue was washed in warm Neurobasal media (Cat# 21103049, Gibco), mechanically dissociated, and strained through a 40 µm cell strainer. The cell suspension was pelleted at $1000 \times g$ for 5 min and resuspended in 2 mL of neuron media (Neurobasal media containing 1% B27, 2 mM GlutaMAX, and penicillin-streptomycin). The neurons were seeded on poly-D-lysine (Cat#P0899, Sigma) coated 96-well culture plates (Cat# 655090, Greiner) at 17,000 cells/well. Cells were maintained at 37 °C and 5% $CO_2$.

## Filament treatment in neurons

Dilutions of tau filaments were calculated from the filament stock so that 20 µL of diluted filament would reach the desired concentration in the added well. Three wells were treated on each plate. To prepare the filaments, stock tau filaments were diluted first in DPBS and sonicated (Diagenode Bioruptor Pico; 10 cycles; setting medium; 30 seconds on, 30 seconds off). Sonicated filaments were then diluted to the calculated concentration in warmed neuron media. 20 µL of the filaments diluted in warmed media were added to the appropriate wells.

## Immunocytochemistry and image analysis

Wells were washed once in PBS, and then 100 µL of 2% HDTA was gently added to all wells and incubated at RT for 10 min. 100 µL of warmed 8% PFA/8% sucrose was gently added to all wells and incubated for 20 min at RT. The plates were then washed in PBS and blocked in 3% BSA in PBS for one h at RT. The primary antibody [T49 (Millipore Sigma, MABN827)] was diluted in 3% BSA in PBS and incubated for 16 h at 4 °C. The plate was then washed in PBS. The secondary antibody (Alexa Fluor 546 Goat Anti-Mouse IgG (H + L)) was diluted in 3% BSA in PBS and incubated for one h at RT, covered to protect from the light. The plate was washed in PBS and 70 µL of PBS with DAPI (0.1 µg/mL) was added to each well. The plate was imaged on the Zeiss CD7 at 20x with a 0.5 optical with 10 ROIs per well. The Zen software was then used to quantify the number of cells in each image using the DAPI signal and the total area of T49 signal per well. The DAPI count and T49 area were averaged across each well. The average T49 area was then divided by the average DAPI count. This value was then normalized to the average value of the AD tau treated wells.

## Immunohistochemistry

Immunohistochemistry was done following an established protocol to stain for phosphorylated forms of tau.[79] Slides were deparaffinized in xylenes and rehydrated in a descending ethanol series. Microwave antigen retrieval with citric acid (Antigen Unmasking Solution, Citric Acid Based, Vector Laboratories #H-3300) was performed. Slides were blocked in 2% FBS in 0.1 M Tris for one h. Primary antibodies (see Supplementary Table 1) were diluted in 2% FBS in 0.1 M Tris incubated in a humidified chamber overnight (16–18 h) at 4 °C. Slides were washed in 0.1 M Tris for 5 min and blocked in 2% FBS in 0.1 M Tris for 5 min. For the AT8 stain, the slides were incubated with a rabbit anti-mouse secondary antibody (ab190481) for 1 h at room temperature. Slides were washed in 0.1 M Tris for 5 min and blocked in 2% FBS in 0.1 M Tris for 5 min, and biotinylated secondary antibodies incubated at RT for 1 hour in a humidified chamber. Slides were then washed in 0.1 M Tris for 5 min and blocked in 2% FBS 0.1 Tris for 5 min. AB solution (VECTASTAIN Elite ABC Kit, Vector Laboratories #PK-6100)

was prepared 15 min in advance and incubated on slides in a humidified chamber at room temperature for 1 hour. Slides were washed in 0.1 M Tris. DAB solution (ImmPACT DAB Peroxidase Substrate, Vector Laboratories #SK-4105) was applied and slides developed for 10 min. Slides were washed in diH2O and briefly counterstained in Harris Hematoxylin (Thermo Fisher, #6765001). Slides were dehydrated in an ascending ethanol series and cleared in xylenes. Slides were coverslipped with Cytoseal 60 (Fisher Scientific, #22-244-256) and dried overnight before imaging at 20x on the Aperio AT2 by Lecia.

## Pathology quantification

AT8 stain was segmented in QuPath and brain sections were registered to the Allen Brain Atlas CCFv3 using a modified QUINT workflow[39,40,80]. Scanned slides were imported into QuPath v.050 (RRID: SCR_018257) for analysis. An RBG image of each selected brain was exported from QuPath as a PNG, down sampled by a factor of 12, to use for spatial registration in QuickNII[81] (RRID:SCR_016854). A pixel classier was applied to the selected brains to detect positive AT8 signal. A classifier with identical settings was applied across all cohorts. Classified pixels were exported for use as the segmentation input in Nutil[82] (RRID: SCR_017183). Brain Images were uploaded into QuickNII[81] for gross alignment to the Allen Brain Atlas CCFv3, and a JSON file was saved for use in VisuAlign (RRID:SCR_017978) for fine alignment. This final alignment was exported as FLAT and PNF files for use in Nutil. Each file was processed through the Nutil software for the quantitative and spatial analysis of AT8 signal in specific regions of the mouse brain. Nutil generated output files containing object area, region area, area occupies, and object counts from each region withing the Allen Brain Atlas CCFv3. Outputs were then processed through N2U (RRID:SCR_024753) to generate anatomical heatmaps and statistical analysis[39,83].

## Statistics

GraphPad Prism version 10.2.3(403) was used for all tests other than brain pathology. One-way Welch ANOVA tests and Dunnett's T3 multiple comparison test were used for primary culture assays. For the western blotting quantification, unpaired t-tests were used to compare the ratio of phosphorylated tau to total tau in the modified AD tau sample to the untreated AD tau sample. In the comparison of fibril width and pitch, unpaired t-tests were used to compare the measurements of the modified or recombinant tau group to the untreated AD tau sample. The linear regression on rankit transformed outcomes that is built into N2U[83] was used to analyze pathology data in mapped brains with second-generation p-values used to determine statistical differences in individual regions.

## Ethical Statement

All housing, breeding, and procedures were performed according to the NIH Guide for the Care and Use of Experimental Animals and approved by the Van Andel Institute Institutional Animal Care and Use Committee (IACUC) under protocol numbers 23-06-015, 23-06-014, 23-10-023.

## Reporting summary

Further information on research design is available in the Nature Portfolio Reporting Summary linked to this article.

## Data availability

All data supporting the results of this study can be found in the article, supplementary, and source data files. The primary data generated in this study have been deposited in the Zenodo database in tabular form[84] (https://doi.org/10.5281/zenodo.15785378). The mass spectrometry proteomics data have been deposited to the ProteomeXchange Consortium via the PRIDE[85] partner repository with the dataset

identifier PXD062343. Cryo-EM images are available at Figshare (https://doi.org/10.6084/m9.figshare.28687256). All cryo-EM density maps have been deposited in the Electron Microscopy Data Bank (EMDB) under the accession codes: EMD-53247 for full-length PAD12 with the Alzheimer's fold, EMD-53463 for 297-391 with the Alzheimer's fold (4a), EMD-53461 for 297-391 assembled in the presence of copper (12a), EMD-53462 for head-to-head fold type one (42a) and EMD-53464 for head-to-head fold type two (42b). Source data are provided with this paper.

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

## Acknowledgements

We would like to thank the patients and families who participated in this research, without whom this study would not have been possible and the Van Andel Institute Brain Bank (RRID:SCR_026035) for providing tissue for this study. We thank the Van Andel Institute Bioinformatics and Biostatistics Core, especially Zachary Madaj, for their assistance with statistical analysis, the Van Andel Institute Pathology and Biorepository Core (RRID:SCR_022912) for their assistance with tissue sectioning, the Institute Optical Imaging Core (RRID:SCR_021968) for their assistance with imaging, the Van Andel Institute Mass Spectrometry Core (RRID:SCR_024903) for their work with characterizing the kinase-treated tau, and the Van Andel Institute Vivarium (RRID:SCR_023211) for caring for animals. We thank Dr. Peter Davies and the Einstein Institute for providing the PHF1 and MC1 antibody. We thank the Electron Microscopy facility of the Medical Research Council (MRC) Laboratory of Molecular Biology for support with cryo-EM; and Jake Grimmett, Toby Darling and Ivan Clayson for support with high-performance computing. We thank Michel Goedert for valuable discussions. This research was funded in part by Ruth L. Kirschstein National Research Service Award F31-AG084199 to A.K. and NIH grant R01-AG077573 to M.X.H. and by the MRC, as part of U.K. Research and Innovation (UKRI) (MC_UP_A025_1013 to S.H.W.S.). Several images were created with BioRender.com.

## Author contributions

A.K. and M.X.H. conceived the project and designed experiments. A.K., S.L., L.B., L.M., C.C., and H.L. performed experiments. J.A.M., A.L., and K.P. provided reagents. A.K., S.L., and M.X.H. developed experimental protocols and analyzed data. S.H.W.S and M.X.H. supervised the project. A.K. and M.X.H. drafted the manuscript. All authors revised and approved the manuscript.

## Competing interests

The authors declare no competing interests.
