## [Transparent Peer Review file · Nature Communications]

Seed structure and phosphorylation in the fuzzy coat impact tau seeding competency

Corresponding Author: Dr Michael Henderson

Version 0:

Reviewer comments:

Reviewer #1

(Remarks to the Author)

This is well conceived and well executed paper with novel findings. Interestingly, the authors themselves, perhaps preemptively, point out the one omission: “the need to compare truncated 297 fibrils to full-length tau fibrils.” While semi-quantitative measures of seeding support their conclusions, the essence of prion-like properties—high fidelity templated recruitment—should be demonstrated. Although they do not show that the seeded tau pathology has the same structure as the seeds themselves, this evidence is necessary for disease-like spread. They claim this will be tested in future studies. Given the unmatched cryo-EM skills in the authors’ lab, they should either include these data or explain why they prefer to break-up the story into smaller publishable units.

It is not clear in the legend what they are showing by the blue dots along the linear sequence in Fig 5 A,D, and E. Probably phosphorylation sites but it should be stated and the residues identified.

It would be informative to see an overlay of the SAPK4 phosphorylation sites with 12 phosphomimetic sites similar to AD tau PAD12 comparison in Fig 8. Although space is a consideration labeling the aa’s would be helpful to the reader in all the structure panels.

The term fuzzy coat is tossed around a lot. Could the authors provide a somewhat more rigorous definition other than not resolved in the cryo. Are there resolution boundaries between the structured region and the “fuzzy coat.”

Reviewer #2

(Remarks to the Author)

The manuscript by Kasen et al. investigates the factors governing the seeding competency of tau fibrils. The authors demonstrate that in vitro-assembled tau fibril structures do not fully recapitulate the seeding capacity of tau fibrils isolated from AD patient brains. Furthermore, the removal or phosphorylation of the fuzzy coat significantly diminishes AD tau seeding activity. Overall, the study underscores the essential roles of both the ordered core structure and the phosphorylated fuzzy coat in modulating the pathological propagation of tau filaments.

The novelty of this work lies in its effort to correlate tau fibril structure—including the fuzzy coat—with pathological outcomes. However, the mechanistic characterization is limited; the authors have only explored the initial properties of the fibrils and the ensuing pathology, while intermediate processes such as internalization, endo-lysosomal escape, and templated fibril assembly remain inadequately investigated. This omission substantially reduces the impact of the study. Moreover, expanding the dataset would further substantiate the conclusions. I offer the following comments and suggestions in the hope of assisting the authors in strengthening this work.

Major concerns:

1. The authors are advised to characterize the intermediate steps leading from fibril formation to pathology. For instance, it is necessary to determine whether seeds with distinct structures exhibit consistent seeding capabilities in vitro; a thioflavin-T (ThT) assay could be employed to assess their templating efficiency. Additionally, the authors should evaluate whether these fibrils possess equivalent cell entry capacities.

2. Previous studies have emphasized the heterogeneity of in vitro fibrils derived from Tau 297-391 (DOI: 10.1073/pnas.2310067120; DOI: 10.1073/pnas.2310067120; DOI: 10.1073/pnas.2310067120). The authors should clarify whether the fibrils used in their experiments are entirely homogeneous.
3. The potential aggregation of seed clumps may influence the internalization efficiency of fibrils by cells; this factor should be accounted for in the experimental design.
4. PAD12 appears to be a promising tau model. Further in vivo characterization of its capacity to induce tau pathology is recommended.
5. The characterization of samples following enzymatic removal of the fuzzy coat is currently insufficient. It is advisable to employ mass spectrometry to identify the predominant molecular weight species in the digested sample and to determine whether the fibril core has been affected.

Minor concerns:

1. In Figure 1(D), Figure 2(C), and other panels, the fluorescent images for the AD Tau group under the 500 nM condition are missing. It is recommended either to include these images for consistency or to annotate that this concentration results in cell death (TOXIC).
2. In Figure 2(A), the protein sequence schematic in the upper section appears unrelated to the lower section and may cause confusion. It is advisable either to remove it or to provide additional explanatory details.
3. In Figure 7(C), the band corresponding to phosphorylated Tau in the western blot for re-phosphorylation detection is indistinct; specific labeling is recommended. Furthermore, the figure does not indicate the loading differences across lanes, rendering the calculation and presentation of the percentage of phosphorylated protein non-intuitive. A clearer description of the quantitative analysis method is needed.
4. The manuscript currently lacks electron microscopy observations of Tau fibrils following re-phosphorylation by SAPK4. Without these observations, it remains unclear whether the re-phosphorylated fibrils maintain the pre-dephosphorylation conformation, raising the possibility that the observed enhanced seeding capability may be attributed to factors other than phosphorylation. Supplementing the study with electron microscopy images of the re-phosphorylated Tau fibrils, comparable to the observations in Figures 5C and 5F, is recommended.
5. In line 91, "treatement" should be corrected to "treatment."
6. In line 117, "phoshomimetic" should be corrected to "phosphomimetic."
7. In line 130, "(citations)" should be replaced with the appropriate references.
8. In line 170, ensure there is a space before "Our observation."
9. In line 277, "prevelent" should be corrected to "prevalent."
10. In line 397 and related sections, "complete protease inhibitor cocktail" should be corrected to "complete protease inhibitor cocktail."

Reviewer #3

(Remarks to the Author)

The manuscript by Kasen et al. investigates how the structural features of tau filaments and phosphorylation in their disordered regions ("fuzzy coat") influence seeding capacity in neurons and mouse models. The study utilizes both biochemical manipulations (pronase digestion, phosphatase/kinase treatments) and cryo-EM validated tau filament preparations to dissect the contributions of core structure and post-translational modifications to tau propagation. The authors employ recombinant phospho-mimetic tau filaments (PAD12) and re-phosphorylation of dephosphorylated AD-tau to demonstrate that phosphorylation of the fuzzy coat contributes significantly to seeding capacity.

The manuscript is well-written. Moreover, in full disclosure, I am not fully qualified to evaluate the conclusions regarding the biology of Tau and cryo-EM. I focused my review on the mass spectrometry-based analysis of protein phosphorylation. Overall, the experiment seems very well designed and executed. My comments are focused on the data analysis.

1) The manuscript includes phosphoproteomics via LC-MS/MS to evaluate phosphorylation status after kinase treatments. However, the analysis appears limited to site identification. It would greatly strengthen the claims if the authors provided quantitative information about phosphosite abundance. As it stands, it is difficult to assess whether SAPK4 sufficiently restores the key phosphorylation landscape of AD tau.

2) Related to the previous, what does this statement mean? "Amino acids with greater than 50% phosphor-occupancy are underlined. Amino acids with greater than 75% phosphor-occupancy are italicized". Is this a quantitative analysis of phosphorylated peptide divided by the unmodified? Those numbers seem to suggest the scoring system for the accurate localization of the phosphorylation on the peptide sequence (something like ptmRS). This would imply that those percentages are not the % of occupancy, but the % of confidence that the phosphorylation site is properly mapped on the peptide sequence.

I checked the raw data in the supplementary table, and there is no quantification for these peptides. Please, revise this section, as it is unclear.

3) The PAD12 construct is convincingly shown to mimic both the structure and function of AD tau filaments. However, a direct phosphoproteomic comparison between AD tau and PAD12 is lacking. Demonstrating a comparable phospho-pattern would strengthen the claim that PAD12 is a faithful mimic, not only structurally but also functionally in terms of PTMs.

Reviewer comments:

Reviewer #1

(Remarks to the Author)

The authors have responded satisfactorily to the criticisms.

Reviewer #2

(Remarks to the Author)

I don't have further question.

Reviewer #3

(Remarks to the Author)

The revised manuscript of Kasen et al. presents a well-executed study on how tau filament structure and phosphorylation in the disordered "fuzzy coat" affect seeding competency. As noted previously, my expertise is in phosphoproteomics, and my comments focus on that aspect. Specifically:

1) The authors clarified their phosphoproteomics approach and reduced their claims, relying on Western blotting to assess kinase activity. While LC-MS/MS quantitation would strengthen the mechanistic conclusions, the current data support the revised interpretations.

2) The confusing use of "phospho-occupancy" has been corrected, and figure legends are now clearer.

3) Supplementary data comparing PAD12, AD tau, and SAPK4-treated tau phosphorylation patterns addresses my previous concern and reinforces the functional mimicry of PAD12.

Overall, the authors have addressed my comments, and I support publication!

We appreciate the reviewers' comments and believe we can sufficiently address their concerns to merit publication in *Nature Communications*. In this response document, we address each of the reviewers' comments in a pointwise manner. Reviewer comments are provided in *italics* and our responses are provided in blue.

Reviewer #1 (Remarks to the Author)

1. *This is well conceived and well executed paper with novel findings. Interestingly, the authors themselves, perhaps pre-emptively, point out the one omission: “the need to compare truncated 297 fibrils to full-length tau fibrils.” While semi-quantitative measures of seeding support their conclusions, the essence of prion-like properties—high fidelity templated recruitment—should be demonstrated. Although they do not show that the seeded tau pathology has the same structure as the seeds themselves, this evidence is necessary for disease-like spread. They claim this will be tested in future studies. Given the unmatched cryo-EM skills in the authors' lab, they should either include these data or explain why they prefer to break-up the story into smaller publishable units.*

Whether or not seeding in cell culture and animal model recapitulates the structure of the seed filament is an open question. We and other groups have shown that *in vitro* amplified filaments fail to retain the core structure of the seed material. Unfortunately, with currently available filament extraction methods, the amount of seeding that occurs in primary neurons is insufficient for cryo-EM structure determination. Therefore, the optimal system to examine amplification fidelity is in mice. However, mice must be aged and extensive optimization will be required to extract sufficient material and determine the structure of tau inclusions in mice. Although this is an important consideration for future studies, faithful replication of seed structure is not a main point of this paper, and therefore the structure determination of seeded aggregates is outside of the scope of the current manuscript.

2. *It is not clear in the legend what they are showing by the blue dots along the linear sequence in Fig 5 A,D, and E. Probably phosphorylation sites but it should be stated and the residues identified.*

Yes, these are phosphorylation sites, the schematics now include labels, specific phosphorylation sites are identified in Fig. 5E where they are analyzed, and the legend of Figure 5 has been updated to reflect this.

3. *It would be informative to see an overlay of the SAPK4 phosphorylation sites with 12 phosphomimetic sites similar to AD tau PAD12 comparison in Fig 8. Although space is a consideration labeling the aa's would be helpful to the reader in all the structure panels.*

This is a great suggestion. We have now created a supplementary figure (Fig. S8) with aligned comparisons of AD tau phosphorylation sites, PAD12 phospho-mimetic sites, and SAPK4-treated tau phospho-sites. The phosphomimetic sites surrounding the AD tau core in PAD12 filaments are sufficient to show near full recapitulation of the seeding of AD tau, while re-

phosphorylation with SAPK4 partially rescues seeding capacity. Taken together, this suggests that the while each cluster of phospho-sites contribute to increasing seeding capacity, phosphorylation both around the S202-T235 sites as well as S396-S404 are necessary to capture full seeding capacity of AD tau.

4. The term fuzzy coat is tossed around a lot. Could the authors provide a somewhat more rigorous definition other than not resolved in the cryo. Are there resolution boundaries between the structured region and the “fuzzy coat.”

Thank you for this point. We have updated the text to clearly define what is considered the “fuzzy coat” (line 62): “the tau “fuzzy coat”, the disordered regions of tau on both the carboxy-terminal and amino-terminal sides of the ordered filament core (aa1–305, aa379-444)..”

Reviewer #2 (Remarks to the Author)

The manuscript by Kasen et al. investigates the factors governing the seeding competency of tau fibrils. The authors demonstrate that in vitro-assembled tau fibril structures do not fully recapitulate the seeding capacity of tau fibrils isolated from AD patient brains. Furthermore, the removal or phosphorylation of the fuzzy coat significantly diminishes AD tau seeding activity. Overall, the study underscores the essential roles of both the ordered core structure and the phosphorylated fuzzy coat in modulating the pathological propagation of tau filaments.

The novelty of this work lies in its effort to correlate tau fibril structure—including the fuzzy coat—with pathological outcomes. However, the mechanistic characterization is limited; the authors have only explored the initial properties of the fibrils and the ensuing pathology, while intermediate processes such as internalization, endo-lysosomal escape, and templated fibril assembly remain inadequately investigated. This omission substantially reduces the impact of the study. Moreover, expanding the dataset would further substantiate the conclusions. I offer the following comments and suggestions in the hope of assisting the authors in strengthening this work.

Major concerns:

1. *The authors are advised to characterize the intermediate steps leading from fibril formation to pathology. For instance, it is necessary to determine whether seeds with distinct structures exhibit consistent seeding capabilities in vitro; a thioflavin-T (ThT) assay could be employed to assess their templating efficiency. Additionally, the authors should evaluate whether these fibrils possess equivalent cell entry capacities.*

We have discussed this point internally, and it is not clear what additional information we will gain from *in vitro* seeding experiments. *In vitro* seeded aggregation is very different from seeding in cells and *in vivo*. Seeded aggregation of tau requires specific buffer conditions that are not physiological to allow tau to unfold from the paper-clip conformation that prevents fibrillization. Under these conditions, it is likely that all tau filaments can likely induce further fibrillization, and it is unclear how we would interpret any potential kinetic differences under these conditions. In contrast, our current studies, seeding in wildtype primary neurons and in mice are more likely to reflect the biological environment and mechanisms likely to influence tau seeding in Alzheimer's disease.

We considered the use of labeled filaments for an internalization assay; however, the truncated filaments are not efficiently labeled with NHS-ester chemistry due to the lack of free lysine groups in the structured 297-391 region. Furthermore, even with efficient labeling, it is likely that the addition of the labeled group would induce a conformational change in the filament because the addition would be inside the core region. The labeling of the PAD12 filaments is possible and has been shown in (Lövestam et al, 2025); however, we did not assess these in our work because of the lack of labeling possible in the truncated filaments. It is possible that the filaments are not equally internalized, which we have addressed in the discussion. Even if this labeling is possible, it would be difficult to interpret the results as the tag may also impact the ability of the cell to internalize the filament. So, while this type of assay could be beneficial, we feel the caveats of generating these filaments and the challenge of interpreting this data outweighs the possible addition of this work.

2. *Previous studies have emphasized the heterogeneity of in vitro fibrils derived from Tau 297-391 (10.1073/pnas.2310067120; DOI: 10.1073/pnas.2310067120; DOI: 10.1073/pnas.2310067120). The authors should clarify whether the fibrils used in their experiments are entirely homogeneous.*

To remove doubts about the structures of the filaments used for seeding in our experiments, we used a fraction of the same samples for cryo-EM structure determination. In the revised manuscript, we have included percentages of filament types represented by each of the cryo-EM structures. The expected structures represented 90% or greater of all filaments for the preparations used in this study. In our experience, protein purity and shaking conditions are crucial for obtaining the correct structure, and other groups have now been able to replicate our results (PMID: 38154792, 36570831, 39841851).

3. *The potential aggregation of seed clumps may influence the internalization efficiency of fibrils by cells; this factor should be accounted for in the experimental design.*

We agree that the clumping of filaments could limit the internalization of the fibrils and have performed experiments to determine whether the size of filaments is within the range that can be internalized by cells. We include this point in the Discussion section (lines 311-315): “In addition, cross-filament interactions of tau could generate larger tau seeds that are not able to be internalized due to size constraints. Previous work has noted that the recombinant filaments made from the truncated tau construct (297-391), such as 4a and 12a, tend to clump together over time, while PAD12 filaments do not. Our filament measurements show some differences, although all filaments had a majority population within the size window expected to be internalized by cells.”

4. *PAD12 appears to be a promising tau model. Further in vivo characterization of its capacity to induce tau pathology is recommended.*

We agree. PAD12 could potentially represent a highly-reproducible method for inducing seeding of tau in mice. Since one of the primary uses for this tau seed could be in wildtype mice, we injected wildtype mice with PAD12 filaments at the same molar concentration as AD tau and aged the mice 3 months post-injection followed by immunohistochemical analysis.

Remarkably, and in contrast to all previous recombinant tau filaments examined, PAD12 filaments induced seeding of tau pathology in wildtype mice as early as 3 months post-injection in similar regions to AD tau. In some regions, like the supramammillary nucleus, the degree of pathology induced by PAD12 filaments is equivalent to AD tau, while dentate gyrus seeding is less than that of AD tau, consistent with primary neuron experiments that show some differences seeding between AD tau and PAD filaments. Overall, the ability for a recombinant tau filament to seed in wildtype mice presents a useful tool for the field in modeling AD-relevant pathology. The new data have been added as panels in Figure 8 and new text has been added to describe these data.

5. The characterization of samples following enzymatic removal of the fuzzy coat is currently insufficient. It is advisable to employ mass spectrometry to identify the predominant molecular weight species in the digested sample and to determine whether the fibril core has been affected.

The use of pronase to cleave the fuzzy coat off tau filaments is a well-established technique in the AD field (PMID:7679073, 3132715, 15196943). Further, tau filaments used for structural studies go through pronase treatment to cleave the fuzzy coat, and this process leaves the AD tau core unperturbed (PMID: 28678775). In our study, we use multiple antibodies and electron microscopy to show that the fuzzy coat has been cleaved but the AD tau core remains intact.

Minor concerns:

1. *In Figure 1(D), Figure 2(C), and other panels, the fluorescent images for the AD Tau group under the 500 nM condition are missing. It is recommended either to include these images for consistency or to annotate that this concentration results in cell death (TOXIC).*

We thank the reviewer for this feedback. We have noted in the text that AD tau was not treated at 500 nM due to this dose being toxic to the neurons (line 98-99). Figure legends have been updated to include “Treatments that resulted in cell death are annotated as TOXIC.”

2. *In Figure 2(A), the protein sequence schematic in the upper section appears unrelated to the lower section and may cause confusion. It is advisable either to remove it or to provide additional explanatory details.*

Thank you for the feedback. We have updated the figure legend to explain that this diagram is related to the colors of the cartoon schematics of the fibril shown on each cryo-EM structure to enable readers to see which tau domains are in the core of each filament.

3. *In Figure 7(C), the band corresponding to phosphorylated Tau in the western blot for re-phosphorylation detection is indistinct; specific labeling is recommended. Furthermore, the figure does not indicate the loading differences across lanes, rendering the calculation and presentation of the percentage of phosphorylated protein non-intuitive. A clearer description of the quantitative analysis method is needed.*

Western blotting of AD tau shows up a smear due to the large number of variable occupancy of post-translational modifications on tau filaments (PMID: 30890929, 29054878, 30890929). For that reason, the whole lane of tau-positive signal is used for quantification. Additional explanation of the quantification method of the western blot to the methods section (lines 468-472): “The ImageLab software was used to calculate the relative phosphorylated tau signal to total tau signal. The whole lane was outlined to detect fluorescent signal for the phosphorylated and total tau channels for each sample. The phosphorylated tau signal was divided by the total tau signal to obtain the phosphorylated to total tau ratio. These ratios were then divided by the ratio of the control sample to obtain the normalized phosphorylated tau to total tau ratio, presented in the graphs.”

4. *The manuscript currently lacks electron microscopy observations of Tau fibrils following re-phosphorylation by SAPK4. Without these observations, it remains unclear whether the re-phosphorylated fibrils maintain the pre-dephosphorylation conformation, raising the possibility that the observed enhanced seeding capability may be attributed to factors other than phosphorylation. Supplementing the study with electron microscopy images of the re-phosphorylated Tau fibrils, comparable to the observations in Figures 5C and 5F, is recommended.*

Thank you for this suggestion. We have conducted the proposed studies, and the characterization of filaments after SAPK4 treatment is now available in Figure S6B. There was no difference in the width or pitch of the SAPK4 treated AD tau from un-treated AD tau fibrils.

5. *In line 91, “treatment” should be corrected to “treatment.”*
This typo has been corrected.
6. *In line 117, “phoshomimetic” should be corrected to “phosphomimetic.”*
This typo has been corrected.
7. *In line 130, “(citations)” should be replaced with the appropriate references.*
The appropriate references have been added to the manuscript.
8. *In line 170, ensure there is a space before “Our observation.”*
The space has been added.
9. *In line 277, “prevelent” should be corrected to “prevalent.”*
The typo has been corrected
10. *In line 397 and related sections, “complete protease inhibitor cocktail” should be corrected to “complete protease inhibitor cocktail.”*
This misspelling of “complete” is because this is a brand name of the reagent.

Reviewer #3 (Remarks to the Author):

The manuscript by Kasen et al. investigates how the structural features of tau filaments and phosphorylation in their disordered regions (“fuzzy coat”) influence seeding capacity in neurons and mouse models. The study utilizes both biochemical manipulations (pronase digestion, phosphatase/kinase treatments) and cryo-EM validated tau filament preparations to dissect the contributions of core structure and post-translational modifications to tau propagation. The authors employ recombinant phospho-mimetic tau filaments (PAD12) and re-phosphorylation of dephosphorylated AD-tau to demonstrate that phosphorylation of the fuzzy coat contributes significantly to seeding capacity.

1. *The manuscript is well-written. Moreover, in full disclosure, I am not fully qualified to evaluate the conclusions regarding the biology of Tau and cryo-EM. I focused my review on the mass*

spectrometry-based analysis of protein phosphorylation. Overall, the experiment seems very well designed and executed. My comments are focused on the data analysis. The manuscript includes phosphoproteomics via LC-MS/MS to evaluate phosphorylation status after kinase treatments. However, the analysis appears limited to site identification. It would greatly strengthen the claims if the authors provided quantitative information about phosphosite abundance. As it stands, it is difficult to assess whether SAPK4 sufficiently restores the key phosphorylation landscape of AD tau.

Thank you for this point. The LC-MS/MS experiment was performed to measure the relative abundance differences between the untreated and SAPK4 treated samples and absolute quantitation was not performed. We have provided data, now Figure S7, showing the relative abundance differences for the sites S202, T205, S396, and S404. The figures provide confidence in the phosphorylation of residues S202, T205, S396, and S404. Panel A displays the relative abundance of the corresponding phosphopeptides in SAPK4 versus control samples. The two isobaric peptides were co-eluted. (a; SpPVVSGDTSPR and SPVVSVDTSPPR, b; SGYSSPGSpPGTPGSR and SGYSSPGSPGTpPGSR). These phosphopeptides were detected exclusively in SAPK4 samples, with precursor mass accuracy within 1 ppm. To confirm the phosphorylation sites, MS2 spectra were manually interpreted, and fragment ions were confidently assigned to support site localization (Panel B).

2. *Related to the previous, what does this statement mean? “Amino acids with greater than 50% phosphor-occupancy are underlined. Amino acids with greater than 75% phosphor-occupancy are italicized”. Is this a quantitative analysis of phosphorylated peptide divided by the unmodified? Those numbers seem to suggest the scoring system for the accurate localization of the phosphorylation on the peptide sequence (something like ptmRS). This would imply that those percentages are not the % of occupancy, but the % of confidence that the phosphorylation site is properly mapped on the peptide sequence. I checked the raw data in the supplementary table, and there is no quantification for these peptides. Please, revise this section, as it is unclear.*

We apologize for the oversight in including this data. We have now included the data in the file in Zenodo. In summary, we aligned the read sequences back to the protein sequence. We then determined the percentage of reads at a specific amino acid with a measured phosphorylation mark, giving the ‘phospho-occupancy’ at a site. This was then mapped to the entire peptide. We have updated the figure legend to clarify this point as it is an estimation of phospho-occupancy rather than a direct measurement.

3. *The PAD12 construct is convincingly shown to mimic both the structure and function of AD tau filaments. However, a direct phosphoproteomic comparison between AD tau and PAD12 is lacking. Demonstrating a comparable phospho-pattern would strengthen the claim that PAD12 is a faithful mimic, not only structurally but also functionally in terms of PTMs.*

PAD12 filaments are from recombinant tau and contain no phosphorylation sites. Instead, they contain 12 phosphomimetic mutations to mimic the charge shift induce by phosphorylation. We have instead provided a diagram showing the mutation sites on the filament against what phosphorylation sites are known to occur in AD tau (Figure 8A) and have now shifted the legend to clarify that these are phosphomimetic sites.